# Deep neural network model of sound localization replicates "what" and "where" representations in auditory cortex

## Abstract

Unlike visual cortex, whether auditory cortex has parallel pathways for sound identification ("what") and localization ("where") is debated. It also lacks a topographic map of auditory space, like the retinotopy in visual cortex. Here, we examined auditory "what" and "where" representations in deep neural network models of sound localization. Surprisingly, the models learned well-separated clusters by sound type, but not by sound level. Sounds were further organized by spectrogram similarity, and these organizations were aligned with human spatial hearing. Sound type also determined whether the representations of sound locations organized into a map: maps were formed in both the horizontal and vertical planes when sounds contained binaural and monaural localization cues that were topographically organized relative to human ears. However, formation of a spatial map did not improve, but rather deteriorated, both the model's and humans' localization accuracy. Together, our model suggests that "what" cannot be dissociated from "where" in auditory cortex. A space map is created by spatially organized localization cues and is unnecessary for auditory cortex.

## 1 Introduction

In the primate visual cortex, information is processed in a hierarchical manner using two parallel pathways (Mishkin et al., 1983): the ventral, or "what" pathway, and the dorsal, or "where" pathway (Fig. 1a). These two pathways are specialized for visual identification/categorization and localization/movement, respectively. The existence of parallel pathways in the auditory cortex (AC) is highly debated, when it was first proposed around the 2000s (Rauschecker & Tian, 2000) (Fig. 1a). Anatomical studies found that caudal and rostral streams of auditory afferents target dorsal and ventral domains in the macaque monkey prefrontal cortex (Romanski et al., 1999). Neurophysiology study in anesthetized macaque also found that single neurons in the caudal AC were highly tuned for sound locations, whereas neurons in the rostral AC are more selective for conspecific vocalization (Tian et al., 2001). Functional magnetic resonance imaging (fMRI) in humans also revealed a similar dichotomy in AC. Behavior studies in cats and humans further show causality of modulation of these two streams in what and where discrimination tasks (Lomber & Malhotra, 2008; Ahveninen et al., 2013). On the other hand, the theory of auditory parallel stream has been argued since it was first proposed (Belin & Zatorre, 2000; Hall, 2003). Multiple pieces of evidence are against the parallel stream hypothesis. First, the distribution of 'what' is everywhere since neurons in both caudal and rostral areas of awake macaque auditory cortex are equally selective for vocalization (Recanzone, 2008; Bizley & Walker, 2009). Second, although neurons in the caudal AC are more selective for sound locations, highly spatially tuned neurons were also identified in the rostral AC (Woods et al., 2006; Remington & Wang, 2019). Third, AC neurons show multiplexed representation of sound features, i.e., where (location) and what (pitch and timbre) information (Bizley et al., 2009; Walker et al., 2011). Last, behavior studies using the advanced optogenetics tool show that inhibiting one area of AC impaired both spatial and non-spatial hearing (Town et al., 2023).

What makes the above arguments of auditory parallel streams more complex is the other mystery about the auditory space map in AC (Fig. 1b). According to the parallel pathway, spatially tuned neurons should form an orderly space map in the dorsal AC. Neurons selective for sound types should form several clusters in the rostral AC, like the ones in the visual ventral pathway (Bao et al.,

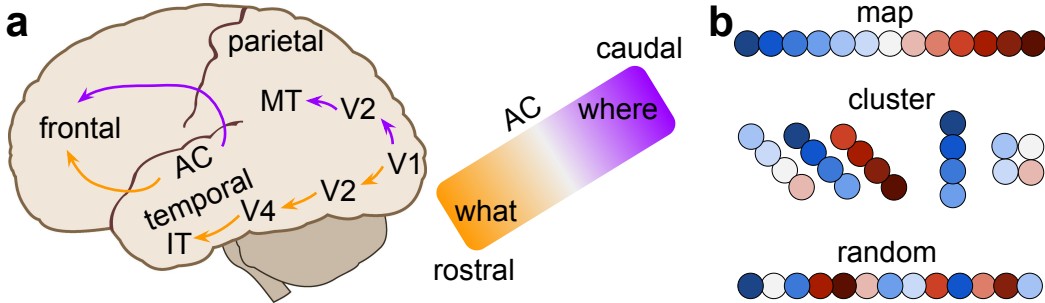

Figure 1: **a**. In visual cortices, the dorsal "where" pathway (purple arrows) starts from the primary visual cortex (V1), and projects to V2 and middle temporal (MT) cortex in the parietal lobe. The ventral "what" pathway (orange arrows) also starts from V1, then projects to V2, V4, and inferior temporal (IT) cortex in the temporal lobe. In the auditory cortex (AC), caudal "where" and rostral "what" pathways project to dorsal and ventral frontal cortex, respectively. **b**. Representations of sound locations (color dots) formed three candidate organizations in the brain. Top: topographic space map. Middle: neighboring neurons on the same electrode from tangential penetration, normal penetration, and from different electrodes but closely spaced normal penetration show similar spatial tuning. Bottom: two-photon calcium imaging (Panniello et al. (2018)) and normal electrode penetration (Remington & Wang (2019)) found neighboring neurons have diverse spatial tunings.

2020). Although there is some evidence of voice patch existence in rostral AC (Petkov et al., 2008), no evidence exists for a space map in any area of AC (Middlebrooks, 2021). A neural map of auditory space was first found in the barn owl (Knudsen & Konishi, 1978), then in the mammalian inferior/superior colliculus (IC/SC) (Chen & Song, 2024; King, 1993). Middlebrooks and colleagues and other groups started to search for such a map in AC since 1981 in cats (Middlebrooks & Pettigrew, 1981). After 40 years of research, based on his and others' work, there is still no evidence for such a map in the AC for all the species that have been examined (Middlebrooks, 2021). Unlike the visual and somatosensory systems with orderly mapped receptors in the retina and skin, the cochlea has a map of sound frequency instead of location. Since the sound locations were computed inside the central nervous system, the representation of those computed but not relayed locations could be in any structure. After two decades of research on the auditory parallel pathway (Rauschecker & Scott, 2009; Recanzone & Cohen, 2010), and four decades of research on the auditory space map, our understanding of two questions is still very limited, partially due to that we lack a computational model for them. There are many models about either only sound spatial and nonspatial attributes. For example, the most famous delay line model (Jeffress, 1948) for computing interaural time difference (ITD), and the spectral-temporal receptive field (STRF) model (Theunissen et al., 2000) for explaining natural sound. Although mechanistic models are useful for explaining how individual neurons are tuned to spatial and nonspatial sound features, they cannot model a spatial organization like parallel streams or a space map. The self-organizing map (SOM) has been used successfully to model orientation and direction maps in the visual cortex (Durbin & Mitchison, 1990). Because their inputs are only handcrafted simple features like line directions, SOM could not take inputs from two sensory attributes or compute the ITD between two audio channels.

## 2 RELATED WORK AND OUR CONTRIBUTIONS

There are several lines of deep neural networks (DNN) modeling works that are related to this study. 1) Visual and auditory 'what' pathway. Using learned features from supervised learning tasks (for example, image recognition or sound classification), encoding models predict, with high accuracy, neural responses in the visual and auditory cortex (Yamins et al., 2014; Li et al., 2023). In particular, Kell et al. (2018) used supervised convolutional neural networks (CNNs) to build encoding models for auditory responses in fMRI recordings and showed an aligned hierarchy between the CNNs and the auditory cortex. 2) Visual 'where' pathway. Like sound frequency, visual locations are faithfully relayed from the retina, thus the visual 'where' pathway's function was mainly on motion processing. Therefore, some studies are focusing on modeling visual motion processing (Güçlü & van Gerven, 2017; Mineault et al., 2021). In particular, Bakhtiari et al. (2021) used self-supervised

learning in a single model with a single loss function to capture the properties of both the visual ventral and the dorsal pathways. 3) Visual 'what' pathway topographical organization. Because standard DNNs have no within-area spatial structure beyond retinotopy, their architecture needs to be modified to model spatial topography. There are two major strategies. One is to add the spatial loss to the task loss (Blauch et al., 2022; Margalit et al., 2024; Deb et al., 2025). The other one is to further train the learnt embeddings using SOM (Doshi & Konkle, 2023; Dehghani et al., 2024; Zhang et al., 2024). 4) Sound localization behaviors. Studies from McDermott lab built DNN models of sound localizations and found they exhibited several characteristics of human psychological behaviors (Francl & McDermott, 2022; Saddler & McDermott, 2024; Banerjee et al., 2025).

We made three contributions in this study.

1) We built the first computational models for both "what" and "where" representations in the auditory system. Our audios contain both "what" (sound types and levels) and "where" (sound locations) attributes of sounds. Our models are hypothesis-free and task-optimized DNN for sound localization only. Because our training data were raw waveforms rather than spectrograms, the model implicitly learned spectrotemporal features while being trained to extract spatial features.

2) We are the first to examine the spatial organization of neural representations from DNN models of the human auditory cortex or human listening task. Previous studies either compare the activations of model and human auditory cortex neurons to the same sound stimuli (Kell et al., 2018; Li et al., 2023; Tuckute et al., 2023), or compare the task performance between models and humans (Saddler et al., 2021; Francl & McDermott, 2022; Feather et al., 2023; Saddler & McDermott, 2024).

3) We are also the first to study the neural mechanisms underlying DNN models of human sound localization behavior. Three previous studies (Francl & McDermott, 2022; Saddler & McDermott, 2024; Banerjee et al., 2025) showed that their models exhibited many features of human spatial hearing based on task performance. By examining the neural representations and their spatial organization, our work fills the gap between DNN models of human behaviors and the auditory cortex.

# 3 RESULTS

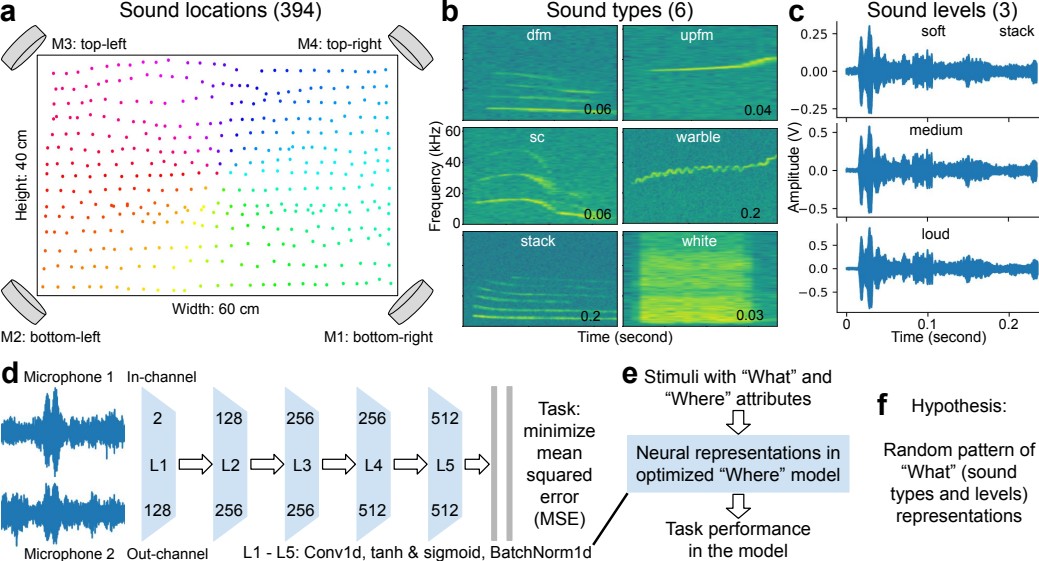

Figure 2: **a**. Four microphones at the corners of the arena will record sounds coming from 394 locations. **b**. Spectrograms of six example sound types. Here, "fm" stands for frequency modulation, "d" for downwards, and "sc" for soft chirp. **c**. Three different sound levels for the call type "stack". **d**. Model architectures. **e**. Pipeline to extract the neural representations. **f**. Our hypothesis.

Here, we modeled the representation of sound locations with a DNN trained for the sound localization task. We used an open-source bioacoustics sound localization dataset (Peterson et al., 2024).

In a 40 by 60-centimeter arena, sounds were played from 394 locations at the bottom (Fig. 2a) and captured by four microphones at the corners that were 35 cm from the bottom. Each location has a median of 144 stimuli, including a fixed number of six sound types and three sound levels, and eight (median) different samples of the same sound type. Sound types include five different classes of gerbil vocalization and one artificial white noise (Fig. 2b). Notice that "dfm", "sc", and "stack" calls have equally spaced frequency bands in their spectrograms. The first or bottom frequency band has a fundamental frequency of f0. Other bands are called harmonics and have a frequency that is an integer multiple of f0. For example, there are three, two, and five harmonics in these three sound types. "upfm" and "warble" only have one narrow band of frequency modulations. In contrast, sound energy was distributed uniformly in the "white" noise call. Notice that the audio waveforms are very different from each other (Supplementary Fig. 1). The medium sound level is calibrated to be approximately the same level as a natural vocalization. The soft and loud sound levels are around 6 dB lower or higher than the medium level (Fig. 2c). We feed two raw audio waveforms to a DNN that has five one-dimensional convolutional layers (Fig. 2d). We trained this model for a sound localization task only. In the trained DNN, we extracted the 512-dimensional embeddings after each layer (Fig. 2e). To visualize the high-dimensional embeddings, we used uniform manifold approximation and projection (UMAP), an unsupervised dimensionality reduction method. Because we only trained this DNN model to localize sound locations ("where"), we hypothesized that the model's representations of sound types and levels ("what") would form random patterns (Fig. 2f).

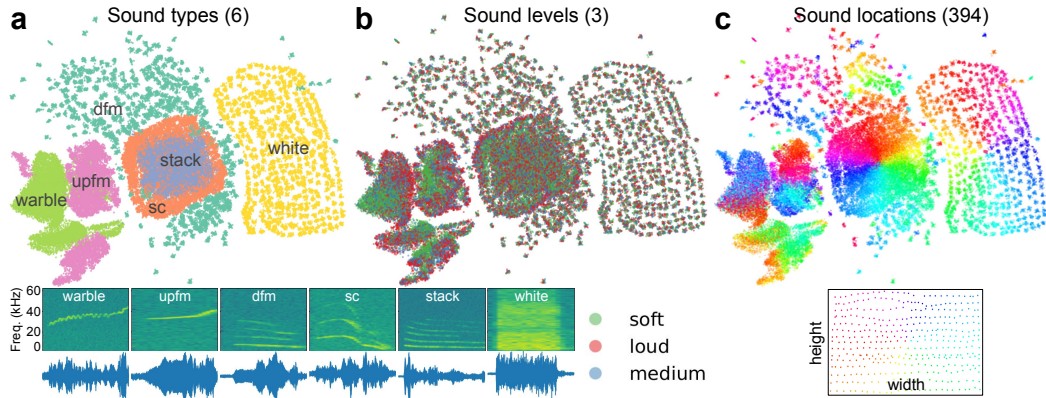

Figure 3: **a-c**. Representations of sound types, sound levels and sound locations.

Surprisingly, this DNN model, which was trained to localize sound locations, formed separate clusters for sound types (Fig. 3a). Clusters from two sound types ("warble" and "upfm") with one narrow frequency band are near each other, and their corresponding representations of sound locations are random (Fig. 3c). Three sound types ("dfm", "stack", and "sc") that all contain multiple harmonics overlapped. The cluster of white noise, which has wide frequency bands, was next to three sound types with multiple frequency bands but far away from sound types with only one frequency band. Its corresponding representations of sound locations form a map (Fig. 3c). Therefore, the DNN model clusters the representations of sound types with similar features. In contrast, the representations of three sound levels were not organized globally (Fig. 3b).

To quantify the differences in clustering between layers, we used the normalized mutual information (NMI). NMI is an information-theoretic metric used to evaluate the similarity between two clusters. The NMI ranges from 0 to 1, with 0 representing no mutual information. Since the representations of sound levels are random globally, their NMI in layers 3 and 5 (Fig. 4b) were both near 0. The NMI for sound type representations decreased from 0.89 in layer 3 to 0.62 in layer 5 (Fig. 4a). This is due to the reorganization of representations for the sound locations (Fig. 4c) because the task is to localize sound locations based on the features extracted from the last layer. Figure 4 shows the representations of all three sound attributes (rows) across all five layers (columns). The sound type representations (Fig. 4a) were not well separated in layer 1 (NMI: 0.54), as two sound types overlapped and all clusters were close to each other. The NMI increased from layer 2 (0.61) to peak at layer 3 and then decreased (0.55). The sound level representations (Fig. 4b) were organized purely according to the sound types or locations. The sound location representations (Fig. 4c) showed the

clearest changes from layer 1 to layer 5. Within layers 1 to 3, the organizations followed the sound type clusters. The "white" noise type of sound began to show a twisted map from layer 2. After layer 3, the representations became scrambled again to better classify sound locations. The map for "white" noise was clear with high spatial resolution. In contrast, the maps for "sc" and "stack" has low resolution. We observed similar patterns in the other two microphones (Supplementary Fig. 2).

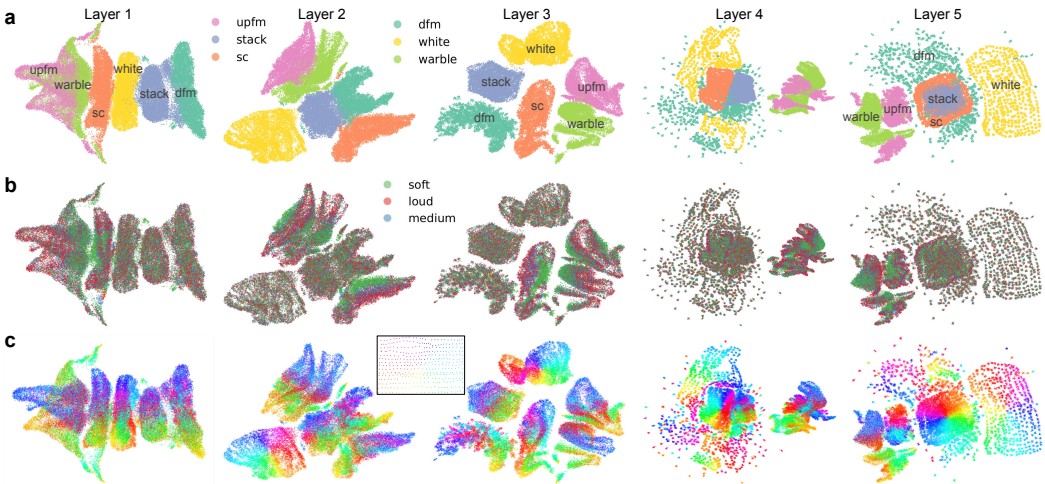

Figure 4: Representations of "what" and "where" attributes of sounds in all layers. **a-c**. Representations of six sound types, three sound levels, and 394 sound locations. Data in layer 5 was the same as Figure 3a-c. The microphone pair is M24. The NMIs for sound types are 0.5454, 0.6140, 0.8898, 0.5488, and 0.6233. The NMIs for sound levels are 0, 0.0001, 0.0025, 0, and 0.

To quantify how well the representation was organized (i.e., organization strength), we used the explained variance R2, which is the coefficient of determination from linear regression between 2D UMAP embeddings and 2D sound locations (height and width). The explained variance of two example sessions was very different (0.99 vs 0.22), with one having a map and the other not (Supplementary Fig. 3a, b). Surprisingly, their sound localization performance (MSE: 0.31 vs 0.37 cm) was similar. Across all six pairs of microphones and five types of stimuli (Supplementary Fig. 3c-e), there was an insignificant ($p = 0.202$) and weak (R2 of X–Y axis = 0.142) correlation between organization strength (Y-axis) and task performance (X-axis). Therefore, a map of sound locations is not always the best choice for representing sound locations. Notice that our model is task-driven and hypothesis-free: we ask it to localize sound locations; we do not impose on it that it create a cluster or map. In summary, a sound-localization task-driven DNN represents sound locations as maps, clusters, or random patterns, similar to those in the superior colliculus and cortices.

We observed consistent findings when using a deeper DNN with ten (instead of five) layers (Supplementary Fig. 4). Two sound types ("upfm" and "warble") that are close to each other and partially overlapping in layer 1 keep this organization all the way to layer 10. In contrast, the other four sound types are only roughly separated in layer 1 and become gradually well separated in deeper layers. The "white" noise type of sound begins to show a spatial map after layer 3 and reaches a peak at layers 6–7. The organization is further refined, with organized clusters formed at each location. The "sc" sound type begins to show maps at layer 7 and reaches its peak at layer 9. All other four types do not show clear organization. We quantified the clustering accuracy (NMI) and organization strength (R2) across all ten layers using different values of UMAP's main hyperparameter, the number of neighbors (Supplementary Fig. 5). Except for the two smallest neighbor values at very deep layers, the NMI values are reliable across choices of neighbors. The R2 scores are always highest for the "white" noise and "sc" sound types and peak in the middle to deep layers, regardless of the number of neighbors. We observed consistent results in the other two microphones as well (Supplementary Fig. 6). Therefore, our findings hold in both shallow and deeper network models.

Our current model takes stereo audio waveforms but does not consider the acoustic cues created by direction-specific filtering of the animal/human body, nor the cochlear properties of the animal/human. Therefore, the models we use here bear little resemblance to the architectures of bi-

ological, especially human, auditory systems. It is also unclear whether our findings (that "what" representations emerge in the "where" model and that a space map is unnecessary) hold for datasets other than gerbil vocalizations. To address the limitations in our models and datasets, we analyzed the neural representations of sound locations in models optimized for human sound localization behavior (Francl & McDermott, 2022; Saddler & McDermott, 2024). These task-optimized models (Supplementary Table 1) received sound inputs that were already filtered by the pinnae, head, and torso (head-related transfer functions, HRTFs), and further filtered by the auditory nerve in cochlea. Since these models exhibit many features of human spatial hearing, analyzing their neural representations can bridge the gap between human sound localization behavior and their auditory systems.

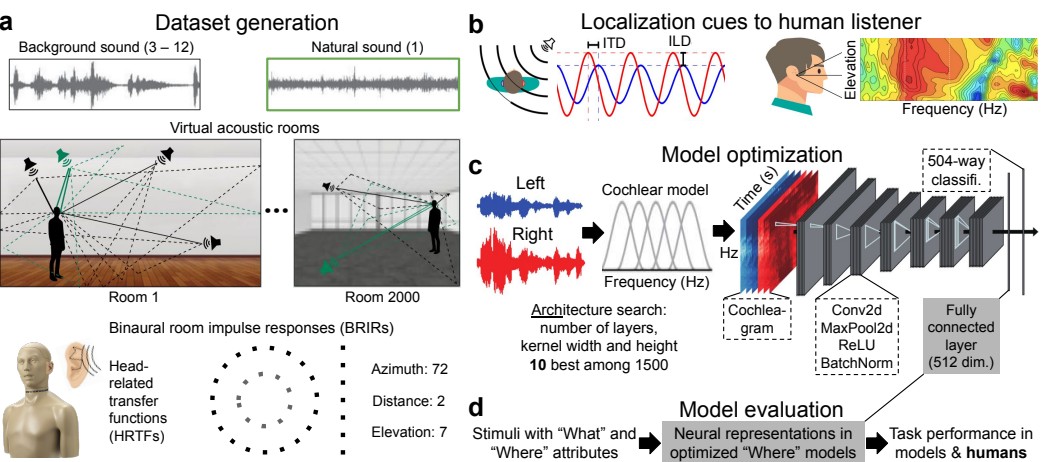

Figure 5: Sound localization dataset and models for human listeners. **a**. Natural sounds (green) are rendered at one location and multiple background sounds (black) are rendered at other locations. There are 2000 different simulated rooms with different sizes and floor and ceiling materials (copied from Fig. 1c of Francl & McDermott (2022)). In addition to the rooms, rendering also includes direction-specific filtering by the head/torso/pinnae, using the HRTFs from the KEMAR manikin (copied from the website of G.R.A.S Acoustics). **b**. Sound localization cues available to human listeners. Left: inter-aural time and level differences (ITDs and ILDs) (copied from Fig. 3b of Saddler & McDermott (2024)). Right: monaural spectral cues to sound elevation. Color-coded HRTFs (amplitude spectra (1–16 kHz) between -15 dB (dark blue) and 20 dB (dark red)) are shown as a function of elevation for a typical human subject (copied from Zonooz et al. (2019)). **c**. Localization model schematic (modified from Fig. 3a of Saddler & McDermott (2024)). **d**. Model evaluation pipeline. This study focuses on the second step (gray shaded area): neural representations.

In real-world listening, there is always noise and reverberation from the environment (Fig. 5a). Saddler & McDermott (2024) simulated a localization-in-noise experiment in which listeners reported which of nine loudspeakers (2 m away, spanning -80° to 80° azimuth in 20° steps) produced a speech utterance, with threshold-equalizing noise played from the remaining eight loudspeakers (top, Fig. 6a). Although the sound azimuth is the same, reverberations blurred the sound waveforms and spectrograms (bottom, Fig. 6a). Therefore, if the sound localization model only needs to segregate the sound locations, then it should be invariant to changes in content, regardless of whether it is clear or blurred. However, this is not the case: representations of sounds at the same locations but with different content are segregated but still connected (dots of the same color with and without black centers, Fig. 6b). Furthermore, the sound azimuths on the left side, right side, and at the midline are also segregated. Our results are consistent across all ten model architectures (Supplementary Fig. 7) and are robust to different hyperparameter values of UMAP in two model architectures (Supplementary Fig. 8). In summary, "where" models learn to represent sound locations by connecting sounds with two different contents at the same location and also represent nearby sound azimuths closer to each other. They form well-segregated clusters (NMI: 1 vs 1, mean, anechoic vs. reverberant) but not a map (R2: 0.607 vs. 0.684). On the other hand, the "where" model also distinguishes the sound contents by forming two segregated but connected clusters.

Saddler & McDermott (2024) showed that precise temporal coding is necessary for sound localization in the horizontal plane. Although we mainly used models with preserved temporal precision

(IHC 3000: inner hair cells with a 3000 Hz low-pass cut-off frequency), we also tested a model with a 50 Hz cut-off frequency (IHC 50) (Fig. 6c, d). Models with lower temporal precision exhibit lower localization accuracy in the reverberant room even with high SNRs (Fig. 6c), but the mechanism is unclear. Our analysis shows that in reverberant conditions, neural representations of sound locations that are far away from the midline overlap (arrows, Fig. 6d). This explains why localization accuracy drops even with high SNRs. This is consistent with human behavior, as human localization is most accurate near the midline (Fig. 3b in Francl & McDermott (2022)). Furthermore, the representations of locations away from the midline are separated instead of connected between anechoic and reverberant conditions (arrowheads, Fig. 6d). Together, reverberation and lower temporal precision affect the spatial organization of representations of sounds away from the midline. Other representation properties are consistent with the high temporal resolution condition.

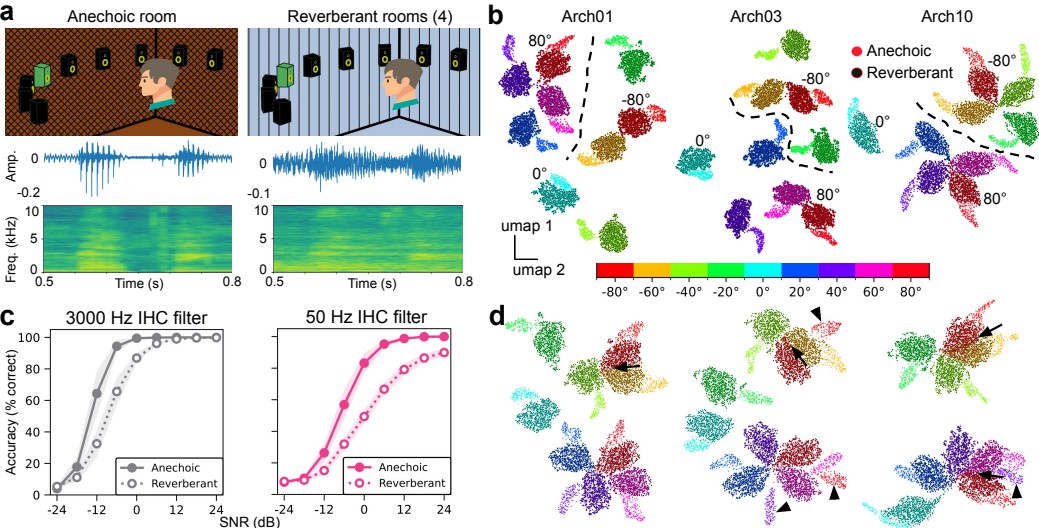

Figure 6: Representations of sound azimuths with reverberation and low temporal precision. **a**. Top: schematic of the sound localization experiment in anechoic and reverberant conditions. There are four types of reverberant rooms and we do not distinguish between them. Bottom: sound waveforms and spectrograms of the same speech segment under the two conditions. **b**. Representations of nine sound azimuths in the two room conditions. Thick dashed lines indicate the boundary between four left and four right sound azimuths. **c**. Model sound localization accuracy as a function of signal-to-noise ratio (SNR) and reverberation. Shaded areas represent the standard deviation across ten model architectures. IHC: inner hair cells. The figures are copied from Fig. 7e in Saddler & McDermott (2024). **d**. Similar to **b** but with a 50 Hz IHC filter. Arrows point to overlapping representations between -80° and -60° or between 60° and 80° in the four reverberant rooms. Arrowheads point to separated representations between anechoic and reverberant rooms at the same azimuths.

The sound contents used in our previous analysis are complex sounds like animal vocalizations and human speech. Although they resemble the real-world sounds heard by animals or humans, they preclude us from explaining our findings. Next, we turned our attention to simple sounds that vary in bandwidth and cut-off frequency in the horizontal (Fig. 7) and vertical (Fig. 8) planes. Saddler & McDermott (2024) compared their task-optimized models with human listeners, which allows us to compare our neural representations to human listeners. In the horizontal plane, human listeners make fewer localization errors when the bandwidth of noise bursts increases (Fig. 7a, b). As shown in Fig. 5b, human listeners mainly rely on binaural ITD and ILD cues for horizontal sound localization. ILDs are significantly affected by the geometry of the head, outer ears, and shoulders, which is why the patterns of the ILDs are much more irregular than the ITDs patterns (Schnupp et al. (2011)). In contrast, ITDs only exist at low frequencies (Brughera et al. (2013)) and are nearly spherically symmetric around the interaural axis (Fig. 7c). Therefore, we hypothesize that low-frequency sounds that preserve ITD cues may help the "where" model form an auditory space map. However, their bandwidth should be neither too narrow (failing to learn any organized representations due to low task performance) nor too broad (producing highly clustered representations for better task performance). Surprisingly, the neural representations confirmed our two hypotheses.

Broader bandwidth noise bursts with higher task performance form well-segregated clusters (dots with black centers, Fig. 7d). In contrast, most, but not all, narrow-band noise bursts form organized space maps. Why only some of them but not all? We found that this is due to the center frequency: high-frequency sounds (over 1.4 kHz, Brughera et al. (2013)) with unmeasurable high thresholds form clusters, whereas low-frequency sounds with low ITD thresholds form maps (Fig. 7e). Our results are highly consistent across all ten model architectures (Supplementary Fig. 9).

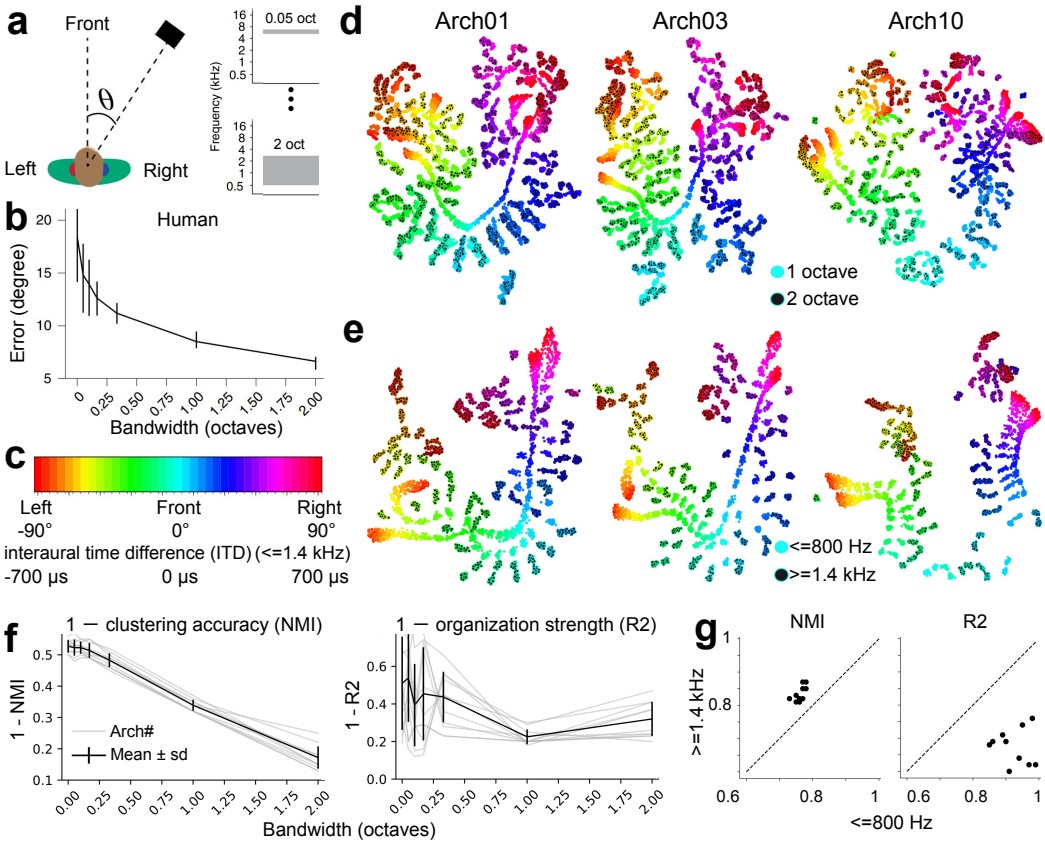

Figure 7: Neural representations of azimuth are bandwidth- and frequency-dependent and aligned with human behaviors. **a, b**. Schematic of stimuli from an experiment measuring the effects of bandwidth and frequency on localization accuracy. Noise bursts varying in bandwidth and frequency are presented at particular azimuth. Human listeners report the azimuthal position with a key press. Error bars indicate the standard deviation across multiple listeners (copied from Fig. 3d of Francl & McDermott (2022)). **c**. HSV color bar that represents sound azimuth from left to right and ITDs from ipsilateral to contralateral. **d, e**. Representations of 37 sound azimuths at two bandwidths and two cut-off frequencies (bandwidth: 1 octave). **f**. The clustering accuracy and organization strength as a function of bandwidth. We used 1-NMI and 1-R2 in order to match the human errors in **b**. Thin gray lines: ten model architectures. Thick black line: mean and one standard deviation. **g**. Scatter plot of NMI and R2 at two cut-off frequencies. Each dot indicates one model architecture.

To quantify our findings and compare them with human behaviors, we plotted the changes in (1 -) clustering accuracy and organization strength against different bandwidths (Fig. 7f). We found that, when increasing the bandwidth in the range where human listeners exhibit low localization error, the clustering accuracy NMI also increased accordingly. Interestingly, the organization strength R2 increased and reached a peak at 1 octave, and then decreased again from 1 to 2 octaves. This reflects the fact that, on one hand, neural representations cannot remain random when task performance is high, and on the other hand, at the highest performance they tend to form clusters rather than a map. Our results are consistent when testing hyper-parameter values in UMAP (Supplementary Fig. 10a). To quantify the bandwidth- and frequency-dependent neural representations, we compared the clustering accuracy and organization strength of all ten model architectures. The NMIs are always

higher when using broader bandwidths or higher center frequencies (Fig. 7g; Supplementary Fig. 10b, c). Together, the datasets with simple sound stimuli show that neural representations in the "where" model are sensitive to simple attributes of sound identity like frequency and bandwidth. Importantly, the formation of a map depends on the available ITD cues but deteriorates the model's and humans' localization accuracy (R2 is highest at 1 octave, but accuracy is lower than at 2 octaves).

Our previous analysis shows that when horizontal sound location cues (ITDs) are spatially organized, their corresponding neural representations also form organized maps. Is this organization specific only to horizontal sound locations? As can be seen in Fig. 8a, b, the HRTFs also vary systematically and monotonically with elevation in the 4–6 kHz band (monotonic region), but their changes are markedly smaller than in the 6–9 kHz band (notch region). By contrast, the cues above 9 kHz seem to be strong but are more erratic (non-monotonic). Therefore, we hypothesized that we would observe a space map at specific spectral cues and bandwidths in the vertical plane.

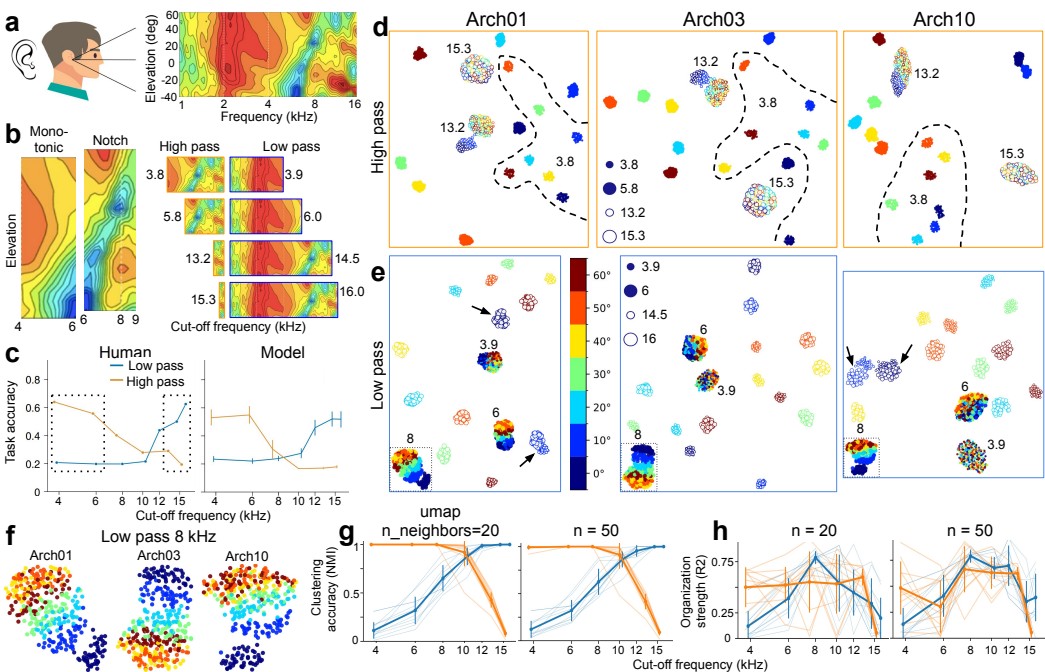

Figure 8: Neural representations of elevation are spectral-cue dependent and aligned with human behaviors. **a**. Spectral cues available to human listeners at different elevations (copied from Fig. 1 of Zonooz et al. (2019)). **b**. Left: two regions in the HRTFs where the spectral cues vary monotonically with elevation. Right: remaining regions in the HRTFs with four different high- and low-pass cut-off frequencies. **c**. Effect of low- and high-pass cut-off frequencies on localization accuracy in humans and models (copied from Fig. 4n, o of Francl & McDermott (2022)). **d**. Representations of seven elevations at four high-pass cut-off frequencies in three model architectures. **e**. Similar to **d** but for the four low-pass cut-off frequencies. Arrows point to overlapping representations from two frequencies at the same elevations. **f**. Elevation maps at an 8 kHz low-pass cut-off frequency. **g**. Clustering accuracy at different high- or low-pass cut-off frequencies with two hyper-parameter values in UMAP. Error bars indicate standard deviation. **h**. Similar to **g** but for organization strength.

Saddler & McDermott (2024) compared their task-optimized models with human listeners at different cut-off frequencies and found that they exhibit similar trends: wider frequency bandwidths lead to higher task accuracy (Fig. 8c). Here, we first examined the representations under high-pass conditions with the two smallest and two largest cut-off frequencies (Fig. 8d). Since only a very narrow bandwidth remained after high-pass filtering at 15.3 kHz, the representations tended to be random and not well separated. The representations tended to form a map at 13.2 kHz (with 0° and 10° separated from the others). In contrast, when broad bandwidths were well preserved, the representations formed seven distinct clusters to represent sound elevations. Importantly, although two sound contents have highly overlapping spectral regions (identical above 5.8 kHz) at the same location, their representations never overlapped in any of the ten models (Supplementary Fig. 11).

With the low-pass condition, the smaller cut-off frequencies resulted in representations that form a random pattern and a roughly organized map, respectively (Fig. 8e). With higher frequencies, the representations of seven elevations form seven distinct clusters. Fig. 8f (also the inset of Fig. 8e) shows the representation at low-pass 8 kHz, where clear space maps are observed. The explanation for this map is the preservation of monotonic and notch regions in the HRTFs, where the spectral cues vary monotonically with elevation. If this is the case, then why does further expanding the bandwidth above 8 kHz make the map disappear? There are two reasons. First, the introduction of extra bandwidth conflicts with the existing monotonic changes. For example, below -20° elevation, the values in the HRTFs are negative around 6 kHz but positive around 12 kHz. The second reason is that forming maps, which causes overlap between nearby elevations, would conflict with the high localization accuracy at broad bandwidths (where forming clusters is the best option).

Figs. 8g, h show the quantified results of clustering accuracy and organization strength at two values of the UMAP hyperparameters. The clustering accuracy NMI is consistent with the task accuracy in the human listener and model, i.e., it monotonically increases and decreases for low- and high-pass cut-off frequencies, respectively. In contrast, the trend of organization strength R2 is different from accuracy, suggesting that whether representations form a map or not is unrelated to accuracy in behavior or in the model. In the low-pass condition, the organization strength peaked at 8 kHz (blue lines, Fig. 8h) and was consistent in all models (Supplementary Fig. 12). Together, the datasets on localization in elevation show that neural representations in the "where" model are sensitive to the available sound content. The formation of a map depends on whether the spectral cues contain only one region that varies monotonically with elevation, and deteriorates the model's and humans' localization accuracy (R2 is highest at low-pass 8 kHz, but accuracy is intermediate).

## 4 DISCUSSION

Here, we examined auditory "what" and "where" representations in deep neural network models trained for sound localization. Conceptually, our findings integrate long-standing "what/where" debates with modern representation learning. We built a task-optimized model that operates directly on raw binaural waveforms, which are closer to biological inputs than spectrograms, while keeping the architecture intentionally simple. This simplicity may not fully capture the complexity of human auditory systems. In contrast, the pretrained DNN models from (Francl & McDermott, 2022; Saddler & McDermott, 2024) were optimized over 1000 architectures, equipped with human ears and cochlea (via HRTFs and auditory-nerve front ends), and trained on human speech, reverberation, and noise bursts of different bandwidths and center frequencies, with spectrograms as model inputs. These models therefore complement our own simplified model, which has no architecture search and is trained on gerbil vocalizations. Thus, our findings are robust across both models and datasets.

The emergence of "what" in "where" streams suggests a way to resolve the apparent conflict between two dual-stream theories of auditory processing: one emphasizing ventral "what" and dorsal "where" streams (Rauschecker & Scott, 2009), and the other focusing on speech processing (Hickok & Poeppel, 2007). In the latter framework, a ventral stream processes speech signals for comprehension, and a dorsal stream (via Wernicke's area) maps acoustic speech signals onto frontal articulatory networks. Our results are consistent with the idea that a dorsal "where" model can carry both the content and the location of speech. Our results do not imply that a separate "what" pathway is unnecessary in auditory cortex. Instead, "what" attributes of sounds such as pitch (Bendor & Wang, 2005; Norman-Haignere et al., 2013), voice (Belin et al., 2000; Petkov et al., 2008), speech and music (Norman-Haignere et al., 2015), and song (Norman-Haignere et al., 2022) do form clusters, mainly in the rostral auditory cortex of primates. If DNN models are trained to perform purely "what" tasks or combined "what" and "where" tasks (Saddler et al., 2025), the representation of "what" should be even clearer than in "where" only task optimized models.

Our results suggest that a space map is created by spatially organized localization cues. This is further supported by maps of sound localization cues in the auditory brainstem (Olsen et al., 1989; Carr & Konishi, 1990). Such maps of localization cues may explain why the cue-independent auditory cortex lacks a map of auditory space (Higgins et al., 2017). We find that the formation of a map does not benefit localization accuracy; instead, it tends to worsen accuracy in both models and human listeners (Fig. 7b vs f; Fig. 8c vs h). The auditory cortex may therefore trade the benefits of forming maps (minimizing wire cost, Chklovskii & Koulakov (2004)) against localization accuracy.

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

# A  APPENDIX

## A.1  SUPPLEMENTARY FIGURES AND TABLE

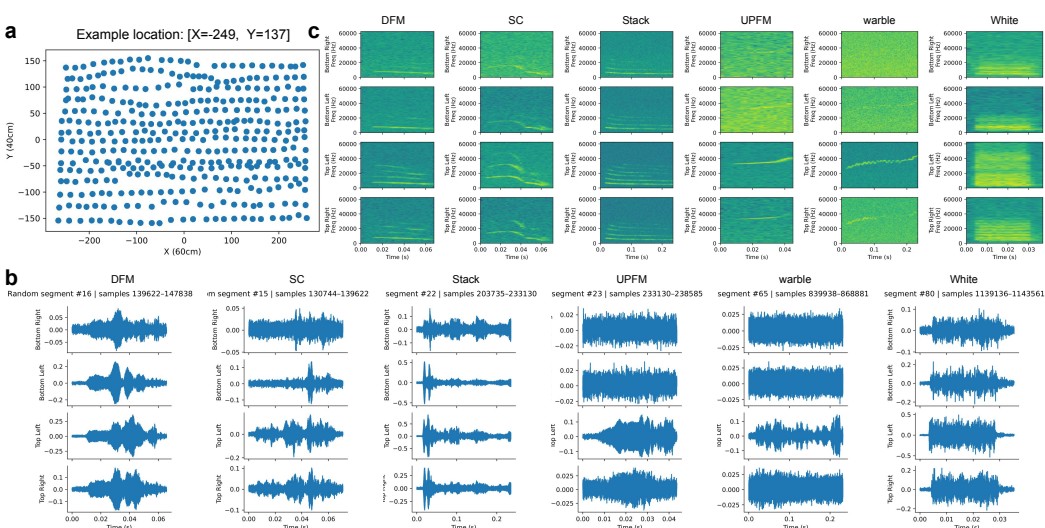

Supplementary Figure 1: Audio waveforms and spectrograms of six sound types from an example location. **a**. There are 394 different sound locations, and the example location comes from the top left corner. Each location (big blue dot) has many (i.e., 144) smaller dots inside since different sound types, levels, and a median of eight different samples are presented from there. **b**. Audio waveforms from six sound types (columns) that were recorded by four microphones at four corners. Notice that the amplitudes are different for each plot. **c**. Corresponding spectrograms for waveforms shown in **b**.

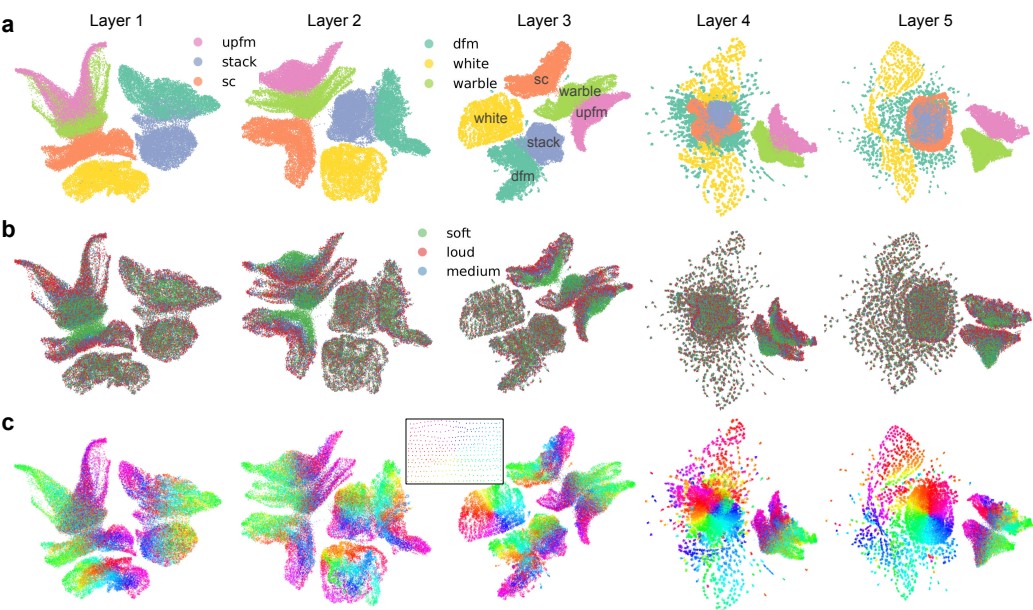

Supplementary Figure 2: Representations of "what" and "where" attributes of sounds in all five layers. Similar to Fig.4, but for the microphone pair of M1 and M3. The NMIs for sound types are 0.6892, 0.8313, 0.8095, 0.4953, and 0.6582. The NMIs for sound levels are 0.0001, 0.0003, 0.0002, 0, and 0.

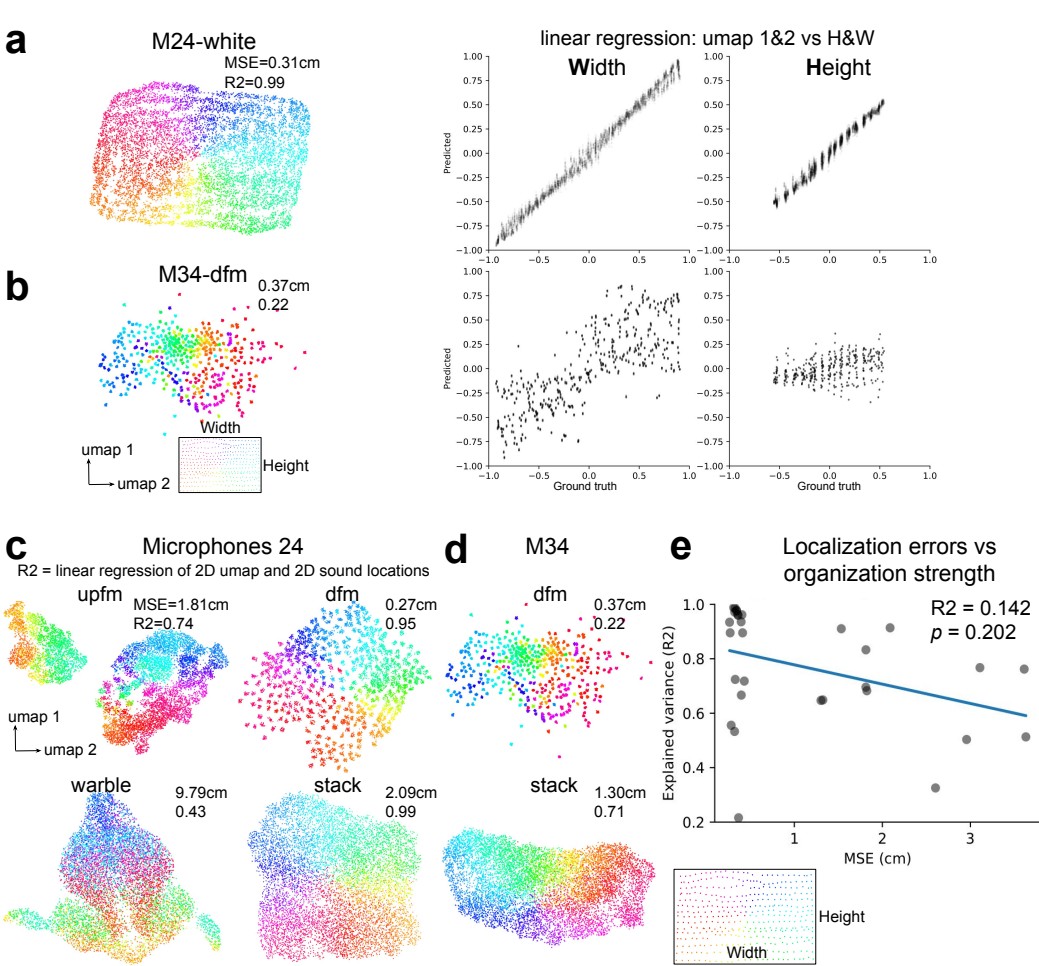

Supplementary Figure 3: Linear regression of sound location representations and locations of the speaker. **a**. A white noise sound type from the microphone pair of M2 and M4. Left, 2D UMAP visualization of sound location representation. Middle, linear regression of 2D UMAP against the width of speaker location. Right, regression of UMAP against the height of speaker location. **b**. Similar to a but for a different microphone pair and sound type. **c**. Same microphone pair as Fig.4c, but sound stimuli from each sound type were trained separately. **d**. Two sound types from a different pair of microphones. **e**. Each dot represents one session (30 total: five sound types × six microphone pairs). Data from the "warble" call type are excluded because their MSE values exceed 10 cm.

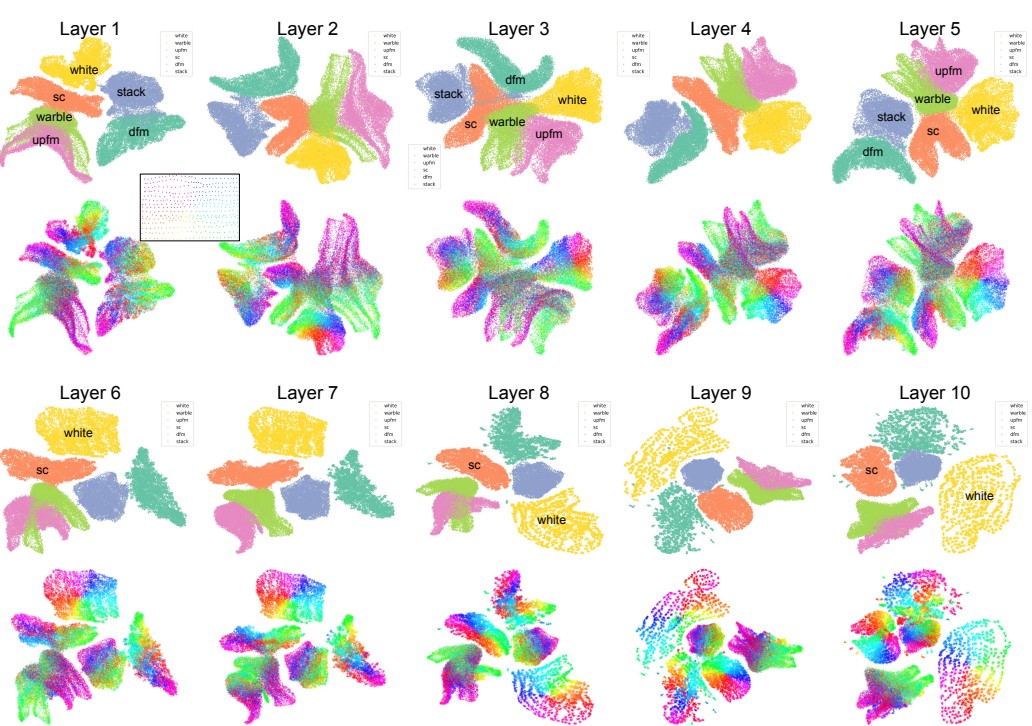

Supplementary Figure 4: Representations of sound types (1st and 3rd rows) and locations (2nd and 4th rows) from layer 1 to layer 10 in a deeper CNN for the microphone pair of M1 and M3. The number of neighbors in UMAP is 200.

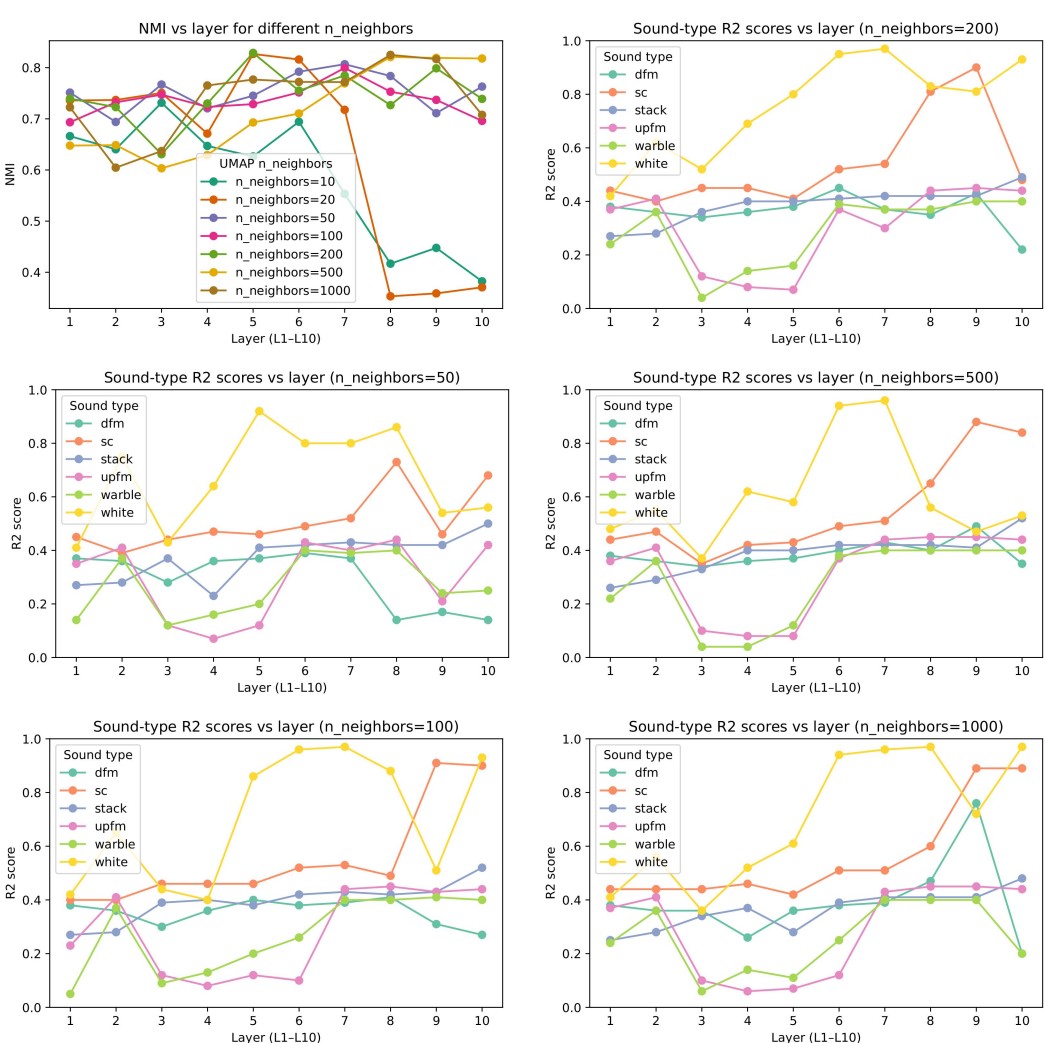

Supplementary Figure 5: The NMIs (top-left) and R2 scores (remaining five panels) across all ten layers. The microphone pair is M1 and M3. The NMIs panel quantifies the clustering performance when changing the hyperparameter (number of neighbors) of UMAP. Notice that the NMIs drop suddenly after layer 6 when the number of neighbors is 10 and 20. Therefore, we only show the R2 scores of six sound types across ten layers with the number of neighbors larger than 20.

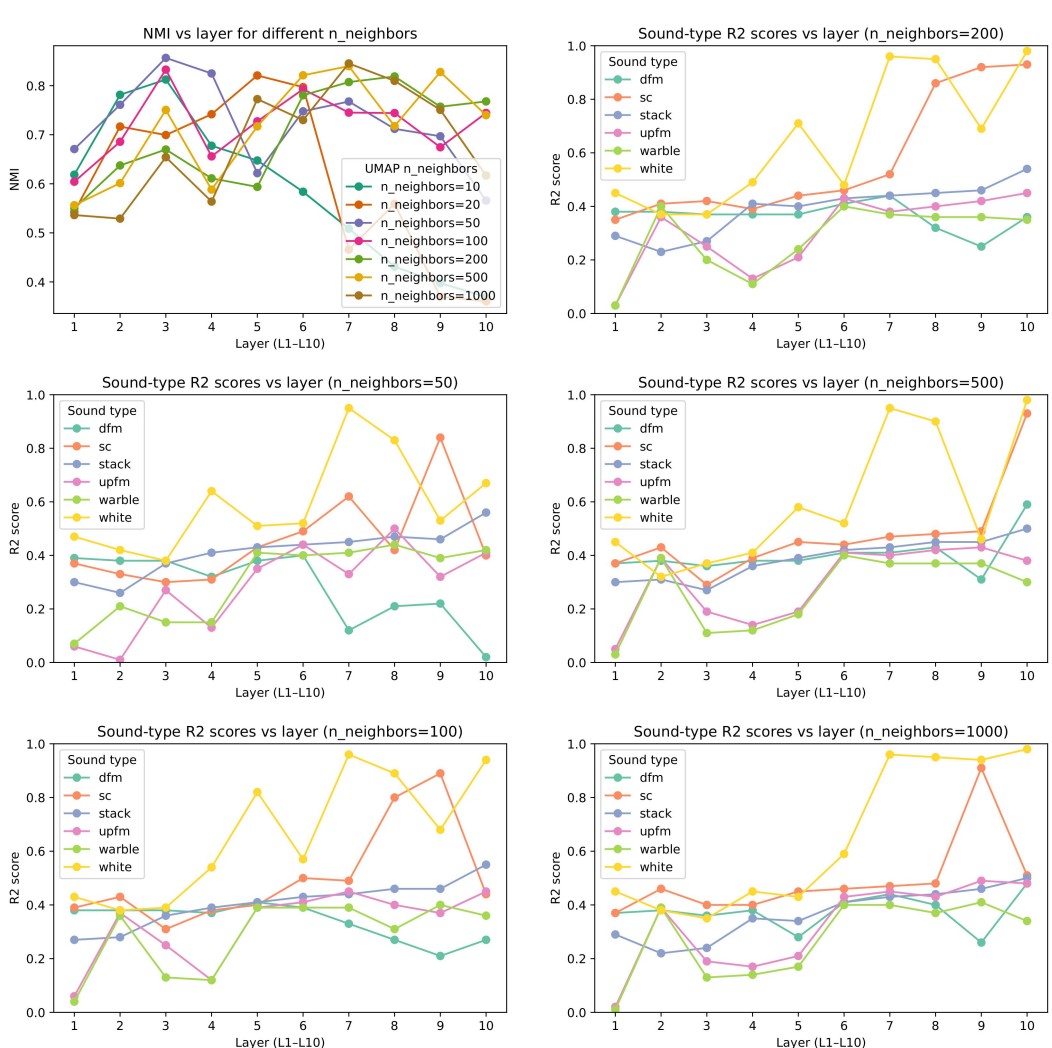

Supplementary Figure 6: Similar to Supplementary Fig.5 but with microphone pair of M2 and M4. Notice that the NMIs also drop suddenly after layer 6 when the number of neighbors is 10 and 20.

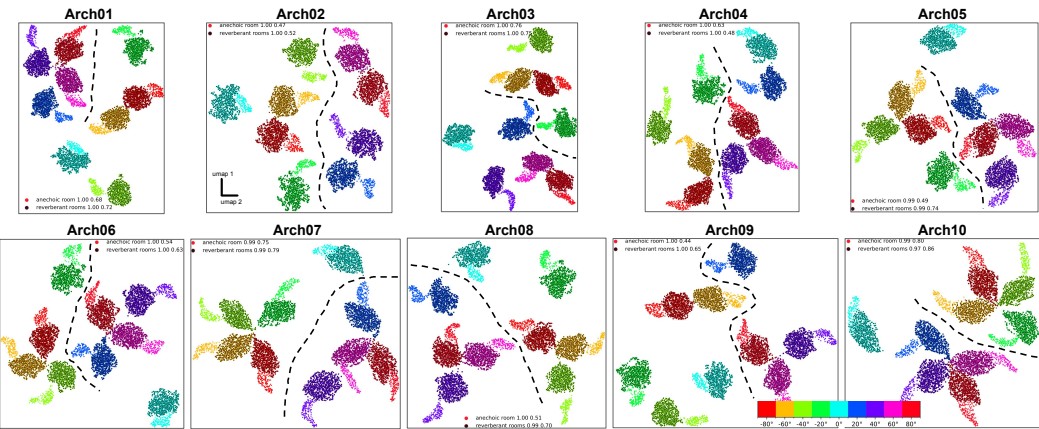

Supplementary Figure 7: Representations of nine sound azimuth angles in the anechoic and reverberant rooms in all ten model architectures. The number of neighbors in UMAP is 20. The dashed black lines indicate the manually drawn boundary between four left and four right azimuths. In each figure legend, the first value is NMI (clustering performance), and the second value is R2-score (organization strength). Three panels (Arch 01, 03, and 10) are shown in Fig. 6b.

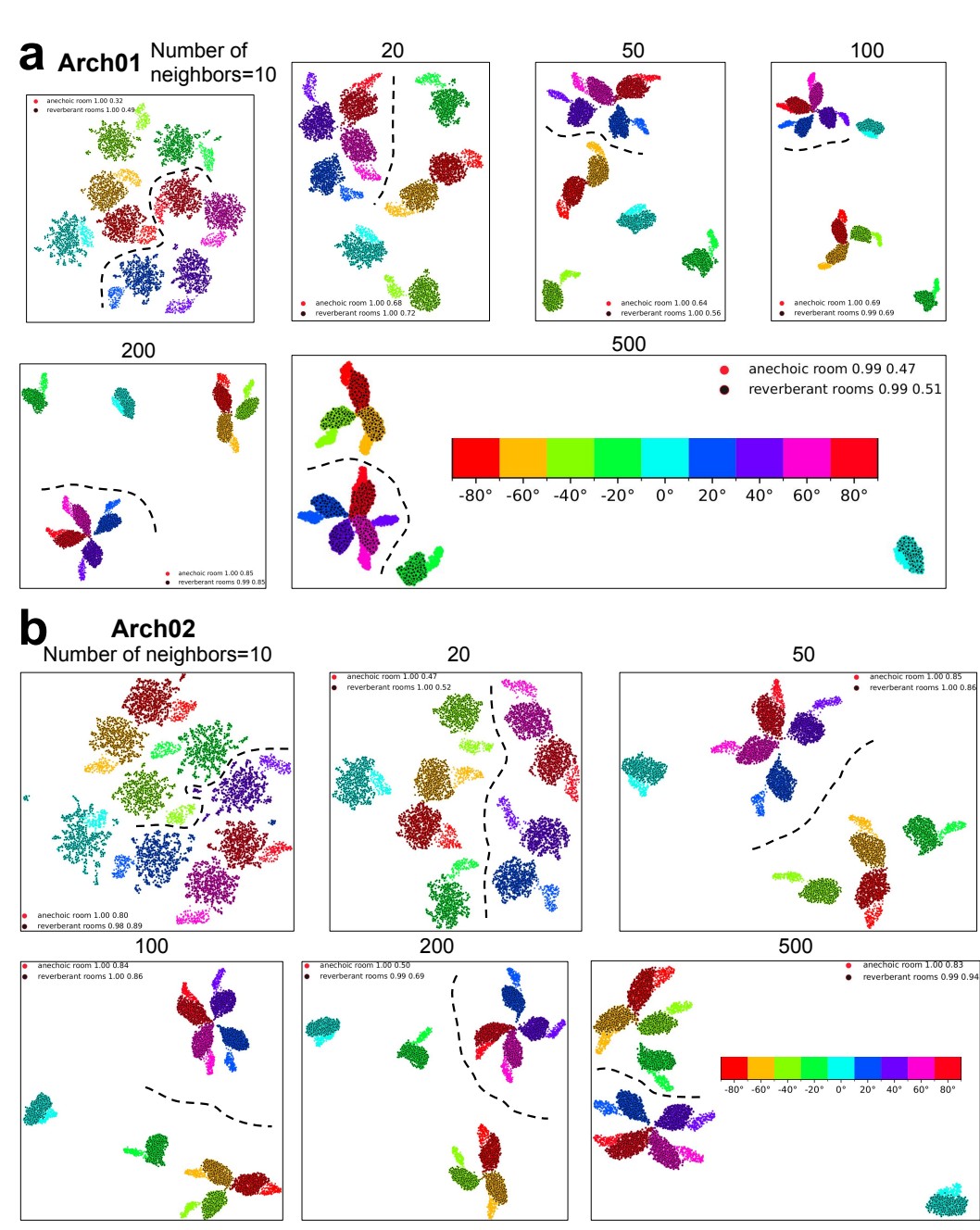

Supplementary Figure 8: **a**. Representations of nine sound azimuth angles in the anechoic and reverberant rooms with six different hyper-parameter (number of neighbors) values of UMAP. The dashed black lines indicate the manually drawn boundary between four left and four right azimuths. In each figure legend, the first value is NMI (clustering performance), and the second value is R2-score (organization strength). One panel (number of neighbors is 20) is shown in Fig. 6b. **b**. Similar to **a** but with model architecture 02.

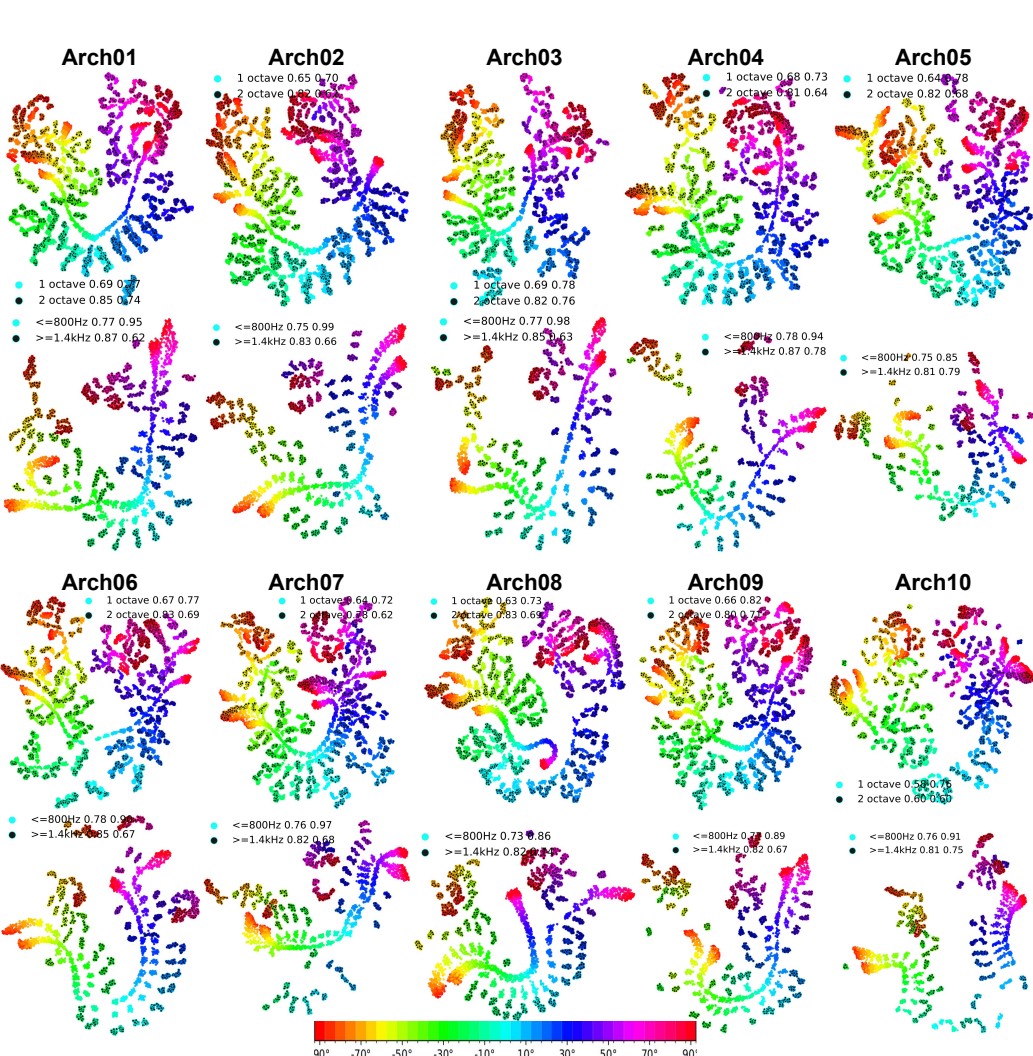

Supplementary Figure 9: Representations of sound azimuths at different bandwidths (1st and 3rd rows), center frequencies (2nd and 4th rows), and network architectures. Representations in Arch 01, 03, and 10 are shown in Fig. 7d, e. In each figure legend, the first value is either bandwidth or cut-off frequency, the second value is NMI (clustering performance), and the third value is R2-score (organization strength).

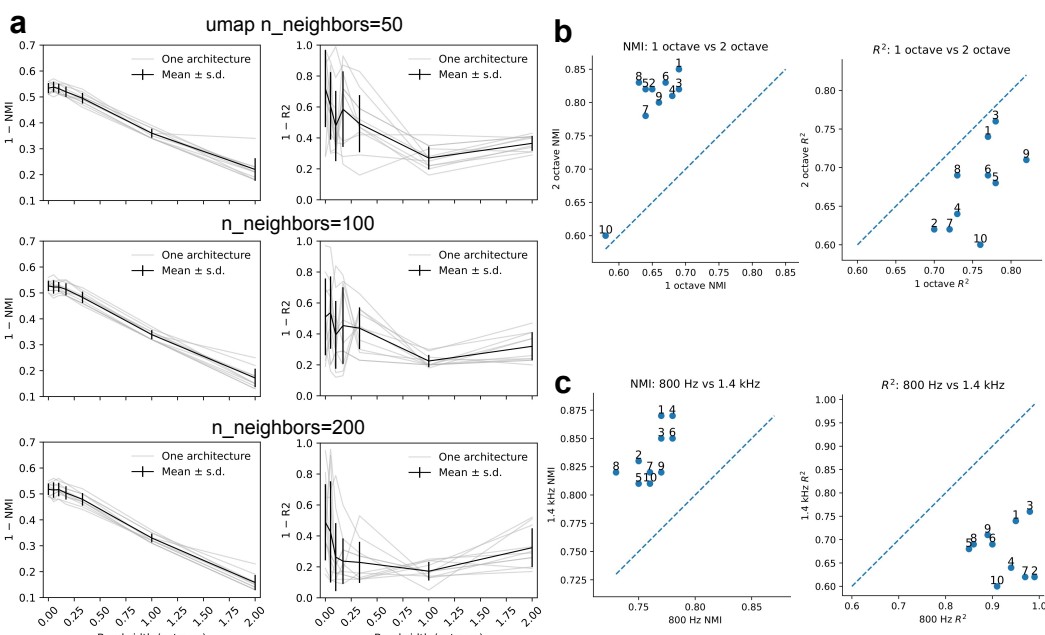

Supplementary Figure 10: The clustering performance and organization strength at different bandwidth, network architectures, and hyperparameter of UMAP. **a**. Similar to Fig. 7f but with three different number of neighbors of UMAP (first to third row). The second row is same to Fig. 7f. **b**. The scatter plot of NMIs (left) and R2 scores (right) across ten model architectures. X-axis is 1 octave bandwidth, and Y-axis is 2 octave bandwidth. The value above each blue dot represents each model architecture. **c**. Same data as Fig. 7g. The scale of X and Y axis for the NMIs panel is different from the R2 scores panel.

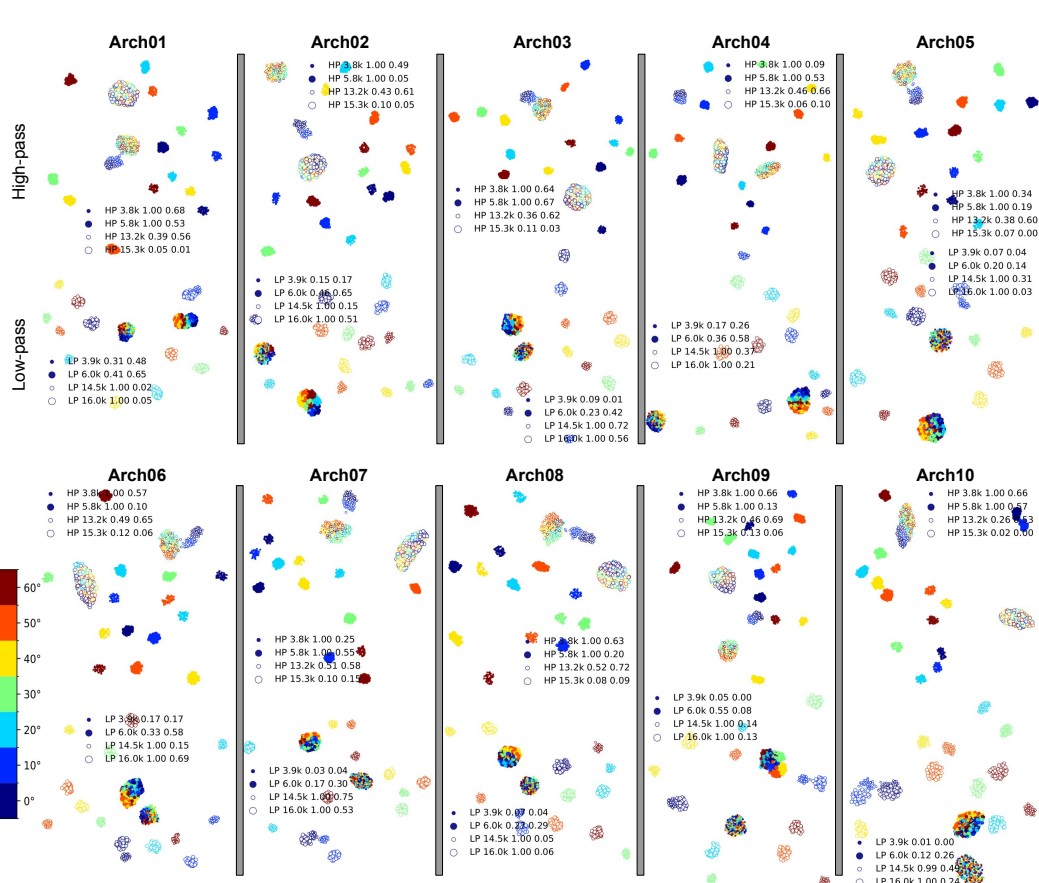

Supplementary Figure 11: Representations of seven sound elevation angles in all ten network architectures at four different high-pass (HP, 1st and 3rd rows) and low-pass (LP, 2nd and 4th rows) cut-off frequencies. Representations in Arch 01, 03, and 10 are shown in Fig. 8d, e. In each figure legend, the first value is cut-off frequency, the second value is NMI (clustering performance), and the third value is R2-score (organization strength).

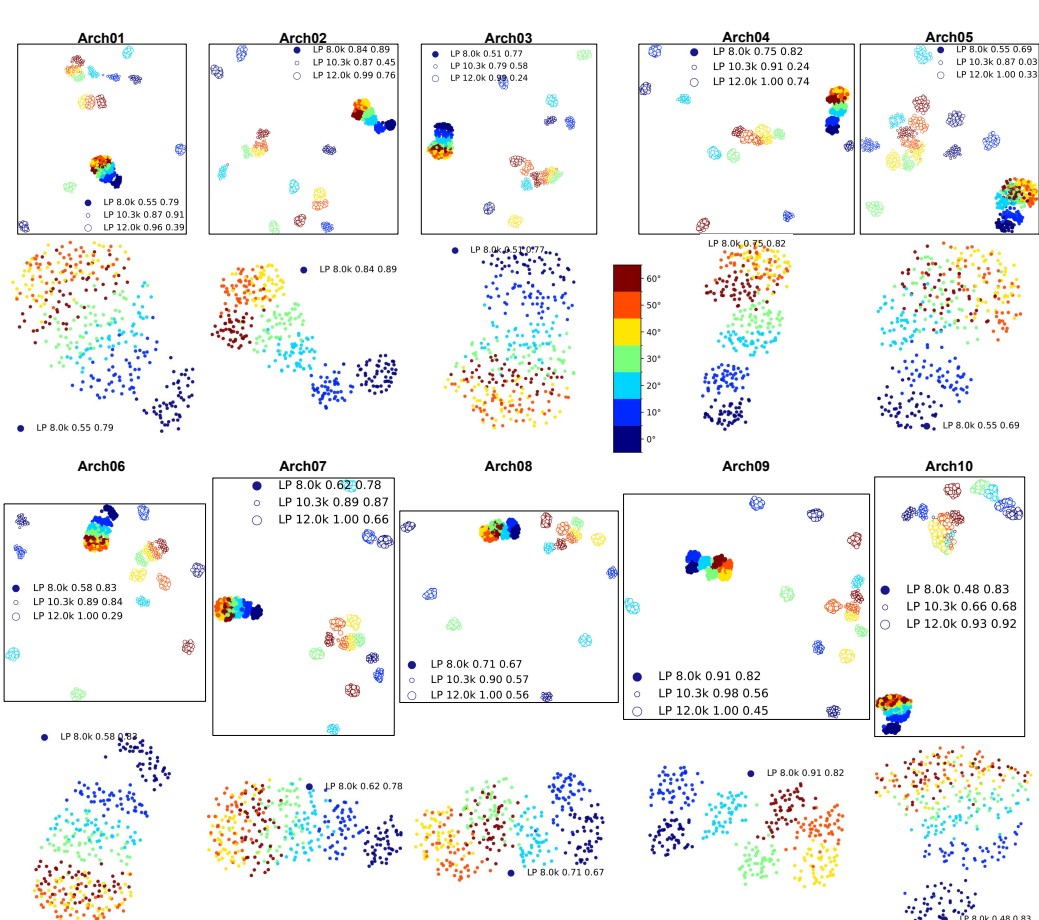

Supplementary Figure 12: Representations of seven sound elevation angles in all ten network architectures at three intermediate low-pass (LP, 1st and 3rd rows) cut-off frequencies. The 2nd and 4th rows show the same representations but with only one 8 kHz low-pass frequency. Representations in Arch 01, 03, and 10 are shown in Fig. 8f. In each figure legend, the first value is low-pass cut-off frequency, the second value is NMI (clustering performance), and the third value is R2-score (organization strength).

| Operation | Network Architecture Numbers | | | | | | | | | |
|---|---|---|---|---|---|---|---|---|---|---|
| | 1 | 2 | 3 | 4 | 5 | 6 | 7 | 8 | 9 | 10 |
| 1 | Conv[1,8,32] | Conv[2,8,32] | Conv[1,4,32] | Conv[3,8,32] | Conv[2,32,32] | Conv[1,64,32] | Conv[1,16,32] | Conv[1,64,32] | Conv[3,32,32] | Conv[2,4,32] |
| 2 | Relu | Relu | Relu | Relu | Pool[1,2] | Pool[1,8] | Relu | Relu | Relu | Pool[2,2] |
| 3 | Bn | Bn | Bn | Bn | Relu | Relu | Bn | Bn | Bn | Relu |
| 4 | Conv[1,64,32] | Conv[3,16,32] | Conv[3,32,32] | Conv[3,8,32] | Bn | Bn | Conv[1,8,32] | Conv[2,16,32] | Conv[2,16,32] | Bn |
| 5 | Relu | Relu | Pool[1,8] | Pool[1,2] | Conv[1,4,64] | Conv[2,4,64] | Pool[1,2] | Pool[1,8] | Pool[1,4] | Conv[2,4,32] |
| 6 | Bn | Bn | Relu | Relu | Pool[1,4] | Relu | Relu | Relu | Relu | Pool[1,4] |
| 7 | Conv[1,64,32] | Conv[2,4,32] | Bn | Bn | Relu | Bn | Bn | Bn | Bn | Relu |
| 8 | Pool[1,8] | Pool[1,8] | Conv[3,32,64] | Conv[1,32,64] | Bn | Conv[1,32,64] | Conv[2,4,64] | Conv[2,4,64] | Conv[2,32,64] | Bn |
| 9 | Relu | Relu | Relu | Relu | Conv[3,2,64] | Pool[2,4] | Relu | Relu | Relu | Conv[3,16,64] |
| 10 | Bn | Bn | Bn | Bn | Relu | Relu | Bn | Bn | Bn | Pool[1,2] |
| 11 | Conv[2,4,64] | Conv[3,16,64] | Conv[1,8,64] | Conv[3,8,64] | Bn | Bn | Conv[2,32,64] | Conv[2,16,64] | Conv[3,4,64] | Relu |
| 12 | Pool[2,4] | Relu | Pool[1,4] | Pool[2,4] | Conv[2,8,64] | Conv[3,4,128] | Pool[1,4] | Relu | Pool[1,4] | Bn |
| 13 | Relu | Bn | Relu | Relu | Relu | Relu | Relu | Bn | Relu | Conv[1,2,128] |
| 14 | Bn | Conv[1,8,64] | Bn | Bn | Bn | Bn | Bn | Conv[1,16,64] | Bn | Pool[1,2] |
| 15 | Conv[3,8,128] | Pool[1,4] | Conv[3,8,64] | Conv[2,2,128] | Conv[1,16,64] | Conv[2,16,128] | Conv[3,2,64] | Pool[1,2] | Conv[3,8,128] | Relu |
| 16 | Relu | Relu | Relu | Pool[1,4] | Pool[1,4] | Pool[1,2] | Relu | Relu | Pool[1,4] | Bn |
| 17 | Bn | Bn | Bn | Relu | Relu | Relu | Bn | Bn | Relu | Fc[512] |
| 18 | Conv[3,32,128] | Conv[3,8,128] | Conv[1,2,64] | Bn | Bn | Bn | Conv[1,2,64] | Conv[2,32,128] | Bn | Relu |
| 19 | Pool[1,4] | Pool[1,4] | Relu | Conv[1,4,256] | Conv[3,4,128] | Conv[1,2,256] | Pool[2,4] | Pool[1,4] | Conv[3,2,256] | Bn |
| 20 | Relu | Relu | Bn | Relu | Pool[1,2] | Relu | Relu | Relu | Pool[1,2] | Dropout |
| 21 | Bn | Bn | Conv[2,2,64] | Bn | Relu | Bn | Bn | Bn | Relu | Out |
| 22 | Conv[3,4,256] | Conv[2,2,128] | Pool[2,4] | Conv[3,2,256] | Bn | Conv[3,4,256] | Conv[1,8,128] | Conv[2,16,128] | Bn | |
| 23 | Relu | Pool[1,2] | Relu | Relu | Conv[3,4,256] | Pool[1,2] | Pool[1,1] | Relu | Conv[2,8,512] | |
| 24 | Bn | Relu | Bn | Bn | Relu | Relu | Relu | Bn | Relu | |
| 25 | Conv[3,8,256] | Bn | Conv[2,4,128] | Conv[2,2,256] | Bn | Bn | Bn | Conv[1,2,128] | Bn | |
| 26 | Pool[1,2] | Conv[3,2,256] | Relu | Pool[1,2] | Conv[3,4,256] | Fc[512] | Fc[512] | Relu | Conv[3,4,512] | |
| 27 | Relu | Pool[1,2] | Bn | Relu | Pool[1,1] | Relu | Relu | Bn | Pool[1,2] | |
| 28 | Bn | Relu | Conv[1,8,128] | Bn | Relu | Bn | Bn | Conv[3,16,128] | Relu | |
| 29 | Fc[512] | Bn | Relu | Fc[512] | Bn | Dropout | Dropout | Pool[1,4] | Bn | |
| 30 | Relu | Conv[1,8,512] | Bn | Relu | Conv[2,4,256] | Out | Out | Relu | Conv[1,3,512] | |
| 31 | Bn | Pool[1,2] | Conv[3,2,128] | Bn | Pool[1,2] | | | Bn | Pool[1,1] | |
| 32 | Dropout | Relu | Pool[1,4] | Dropout | Relu | | | Fc[512] | Relu | |
| 33 | Out | Bn | Relu | Out | Bn | | | Relu | Bn | |
| 34 | | Fc[512] | Bn | | Fc[512] | | | Bn | Fc[512] | |
| 35 | | Relu | Fc[512] | | Relu | | | Dropout | Relu | |
| 36 | | Bn | Relu | | Bn | | | Out | Bn | |
| 37 | | Dropout | Bn | | Dropout | | | | Dropout | |
| 38 | | Out | Dropout | | Out | | | | Out | |
| 39 | | | Out | | | | | | | |

Supplementary Table 1. Summary of the 10 network architectures. Conv[X, Y, Z]: Convolutional layer with kernel height X, kernel width Y, and number of filters Z. ReLu: Rectified linear unit layer. Bn: Batch normalization layer. Pool[X, Y]: Max pooling layer with kernel height X and kernel width Y. Fc[X]: Fully connected layer with X number of units. Dropout: Dropout layer. Out: Softmax classification layer with 504 output units. This table is copied from the Extended Data Fig. 3 of Francl & McDermott (2022) (same as Supplementary Table 1 of Saddler & McDermott (2024)).

## A.2 METHOD

### A.2.1 GERBIL DATASETS AND MODELS

We used the Speaker Dataset (Speaker-4M-E1, Peterson et al. (2024)) which was generated by repeatedly presenting five characteristic gerbil vocal calls and a white noise stimulus at three volume levels (18 total stimulus classes) through an overhead tweeter speaker. Between every set of presentations, the speaker was manually shifted by two centimeters to trace a grid of roughly 400 points along the cage floor. This procedure yielded a dataset of 70,914 presentations spanning the 18 stimulus classes. Gerbil vocalizations can range in frequency from approximately 0.5–60 kHz, and different vocalizations correspond to different types of social interactions in nature. In this study, a diverse set of commonly used vocal types was selected that vary in frequency range and ethological meaning. Data is available at: vclbenchmark.flatironinstitute.org.

The network consists of 1D convolutional blocks connected in series (Fig. 2d). It takes raw multi-channel audio waveforms as input and outputs the mean and covariance of a 2D Gaussian distribution over the environment. We used two model architectures in this study. One is the simple five-layer DNN model shown in Fig. 2d. The other one is the deeper ten-layer DNN model similar to the DNN model used in Peterson et al. (2024). The size of input channels are 2, 128, 128, 128, 128, 256, 256, 256, 256, and 512. The size of output channels are 128, 128, 128, 128, 256, 256, 256, 256,512, and 512. The kernel size is fixed to 33, and the stride is 1, 2, 1, 2, 1, 2, 1, 2, 1, and 2. We extracted the embeddings of each sound stimuli (i.e., neural representations) from different layers at the best epoch (minimal validation loss).

### A.2.2 Human datasets and models

Francl & McDermott (2022) conducted an architecture search over 1500 models and selected the top ten with the lowest validation loss. These models classified noisy 1-second auditory scenes according to the azimuth and elevation of a target natural sound. Source location classes spanned 360° in azimuth (5° bin width) and 0–60° in elevation (10° bin width), yielding a total of 504 output classes (72 azimuth by 7 elevation classes). To ensure that the task was well defined, the training scenes always contained a single natural sound rendered at one target location, superimposed on real-world noise textures diffusely localized at 3–12 different distractor locations. Target sounds were drawn from the Glasgow Isolated Sound Events (GISE-51, Yadav & Foster (2021)) subset of the Freesound Dataset 50k (FSD50K, Fonseca et al. (2021)), which consists of variable-length recordings of individual sources spanning 51 categories of everyday sounds.

To spatialize scenes, Saddler & McDermott (2024) used a virtual acoustic room simulator (Shinn-Cunningham et al., 2001) to render sets of binaural room impulse responses (BRIRs) for a KEMAR manikin in 2000 unique listener environments. The simulator used the image-source method and incorporated KEMAR's HRTFs (Gardner & Martin, 1995). They randomly generated 2000 unique listener environments by sampling different shoebox rooms (varying in size and wall materials) and listener positions (x, y, z coordinates and head angle) within each room. Room lengths and widths were sampled log-uniformly between 3 and 30 m, and room heights were sampled log-uniformly between 2.2 and 10 m. The listener's head position was sampled uniformly within each room, with the constraints that the head was at least 1.45 m from every wall and no higher than 2 m above the floor.

For each listener environment, they rendered BRIRs at 1008 source locations (2 distances from the listener, 72 azimuths, 7 elevations). One of the distances was fixed at 1.4 m for every BRIR, and the other distance was independently sampled for each BRIR (drawn uniformly between 1 m and 0.1 m less than the distance from the listener to the nearest wall). A total of 1800 unique listener environments were included in the training set, and the remaining 200 were used for validation. The final training and validation datasets consisted of 1,814,400 and 201,600 binaural auditory scenes, respectively. Target natural sounds were placed once at each of the 2000 × 1008 source locations to ensure that the dataset was balanced across the 504 target location classes. Auditory scenes were presented to the model during training at sound levels drawn uniformly between 30 and 90 dB (sound pressure level, SPL).

We downloaded the open-source model weights and sound stimuli (https://github.com/msaddler/phaselocknet_torch) from the authors' provided Google Drive link. We used the pretrained models in PyTorch versions (simplified IHC 3000 Hz). We used IHC 50 Hz models only in Fig. 6. Since the pretrained models are only provided in TensorFlow version, we converted them to PyTorch version with their provided code. We used simplified cochlear model because state-of-the-art cochlear models that best capture the response properties of the auditory nerve are computationally expensive (12 TB). Saddler & McDermott (2024) found that greatly simplified cochlear stage qualitatively and in most cases quantitatively replicated the results obtained with the highly detailed model of the auditory nerve.

In each category of models (simplified or not, IHC 50/320/1000/3000 Hz), there are ten model architectures (arch 01 to arch 10, Supplementary Table 1). We showed the results of architecture 01, 03, and 10 in the Figs. 678 since they represent CNN with medium (8), deep (10), and shallow (4) layers (Supplementary Table 1). We also showed the results of all ten model architectures in the Supplementary Figs. Those ten models are evaluated on ten different sound localization environments and we choose three representative experiments: anechoic vs reverberation (Fig. 6) in the horizontal direction, and bandwidth and frequency dependent sound localization in the horizontal (Fig. 7) and vertical (Fig. 8) directions.

We used the sound stimuli from "evaluation" datasets (363 GB), and we used three out of ten datasets: speech_in_noise_in_reverb (8.6 GB), bandwidth_dependency (19.6 GB), and mp_spectral_cues (6.9 GB).

Figure 6: anechoic vs reverberation. There are 18,800 stimuli, including 376 speech excerpts, 10 SNRs (-24, -18, -12, -6, 0, 6, 12, 18, 24 dB, plus inf), and 5 rooms ("index room": 0, 1, 2, 3, 4). The index room = 1 was anechoic room without reverberations. The index room = 0 was the reverberant room Saddler & McDermott (2024) featured in the paper. The remaining three reverberant rooms

(2, 3, 4) have low, medium, and high levels of reverberation, but did not use authors in the original paper. We used all four reverberant rooms in this study, therefore there are four times of data points in reverberant rooms than anechoic room (Fig. 6b, d).

Figure 7: bandwidth and frequency in the horizontal plane There are 55,500 stimuli, including 37 azimuths (-90° to 90° in steps of 5°, 0° elevation), 12 bandwidths (we only analyzed seven of them to match the human experiments: 0, 1/20, 1/10, 1/6, 1/3, 1, 2), and 125 frequencies (either pure tone or center-frequency of noise burst).

Figure 8: Low-pass and high-pass cut-off frequency of spectral cues There are 9,800 stimuli, including 2 azimuths (front and rear midline), 7 elevations (0° to 60° in steps 10°), 700 combinations of different low-pass (3.9, 6.0, 8.0, 10.3, 12.0, 14.5 or 16.0 kHz) and high-pass (3.8, 5.8, 7.5, 10.0, 13.2 or 15.3 kHz) cut-off frequencies.

We extracted the embeddings of sound stimuli after passing them through each pretrained model. In each simplified version of DNN model, it contains two parts: one is Peripheral Model which contains Gammatone filter bank to process the audio inputs, and it is same among ten model architectures. The other one is Perceptual Model which is different model architectures. We extract the embeddings (neural representations, 512 dimensions) of each sounds after multiple convolutional layers (depending on model architectures) at the "fc_intermediate" layer.