# OpenReview forum: "Deep neural network model of sound localization replicates “what” and “where” representations in auditory cortex"
_ICLR.cc/2026/Conference — Submitted to ICLR 2026_

### Official Review · Reviewer_i53z · 2025-10-28

**Soundness:** 3
**Presentation:** 4
**Contribution:** 2
**Rating:** 2
**Confidence:** 4

**Summary:**

In this work, the Authors invesigate the co-localization of spatial and contentual representations in auditory processing. Motivated by the question of how these representations are processed in the brain, they trained a neural network where they investiigated the relations between the representations of the content and location in the auditory processing. As a result, they provide a comprehensive display of these representations on different stages of auditory processing in their model.

**Strengths:**

- Thorough literature review

The paper offers a substantial literature review that clearly motivates the problem at hand and provides an account of the related work on it. Different theories / models are mentioned and their support / valifdity is thoroughly discussed. This offers a nice introdcution into the topic and the related research, clearly builing up to the gap that this work aims to address.

- Clear text, high-quality figures.

The text overall is well-structured. It clearly devivers the message of the work and substantially describes the steps taken within the research project, paired with the corresponding results. The figures are clear and intuitive, showing the work's results chiefly throug the visualized dimensionality reduction / clustering techniques, easy to visualize parse. The font are of human-readable size, which is rare among the submissions here.

- Well-founded approach

I am personally a big fan of this approach to understanding and interpreting the representations learned by deep-learning models on various levels. In auditory processing, this approach has been highly successful in the past: Applied to Transformer models trained for speech recognition (Toyota institute circa 2019), it helped determining which layers of the Transformer stack encode syllables, words, sentences, and semantic content shich, in turn, has allowed to optimize the models. Naturally, the same approach can be applied to the broader spectrum of auditory tasks, as done in this paper.

- Comprehensive investigation

As represented in the many figures of the paper, the representations of the sound have been compared to a large set of variables, resulting in the comprehensive investigation of the representations learned by the model.

**Weaknesses:**

While I find the overall approach taken in this paper to be proper and interesting, there are two considerations regarding to the way it's used here.

- Limited to one network architecture

While comprehensive, all the analysis is performed for a single architecture trained in a single way. While this approach is shown to be successfull when the goal is to understand and improve a specific model architecture (i.e. model-centric), it's utility is less clear if the goal is to gain general knowledge regarding the auditory processing or the auditory system in the brain. The identified representations may be specific to the proposed architecture, data, and trining regime of the model. While the results acquired this way can still serve as a proof of principle (e.g. that mixed representations of "where" and "what" are plausible in auditory decoding models), furhter considerations are needed to put out more general claims.

- Parallels to known properties of the auditory system are not explicitly discussed.

One way to tie the results to the auditory system in the brain would be to explicitly discsuss what properties of the auditory processing, as observed perviously in the brain, were reprodiuced by the model. While the Authors have provided us with the comprehensive introduction that lists such properties, the paper could benefit from an explicit discussion section detailing which of these properties have been observed in the model.

Overall, these two points limit the scope of the claims that can be made in this work. While the Contributions section of the paper does not overstep these boundaries, the claims are mainly specific / limited to the considered model without further ties to the auditory system, thus not necessarily substantiating the claim in the paper's title.

**Questions:**

Could you please futher comment on the scope of the findings, i.e. on the way how they relate to the auditory processing in the brain? Are there any further parallels that can be made of any further implications that can be derived?

---

> ### Author Response · Authors · 2025-11-27
> **One architecture and parallels to auditory system**
>
> We appreciate this reviewer for the highly positive comments on the "Presentation" of our work. We hope that our revised manuscript also satisfies this standard. We understand the reviewer’s concern about the "Contribution" of our work. In our revised manuscript, we added a ten-layer model (L254–267, blue fonts; Supplementary Figs. 4–6) that accompanies our previous five-layer model. This is of course not enough by itself, but we believe our human results will definitely resolve the concern about model architecture. Adding the human results also allows us to tie our findings to the human behaviors for the first time (see our Contributions section in L117–133). For example, the clustering accuracy of neural representations is correlated with the models’ and humans’ accuracy, but organization strength is not.
>
> We cannot quantitatively relate our results to auditory system due to the lack of open-source experimental data. However, in the last paragraph of the revised Discussion, we do mention parallels to known properties of auditory system (L534–539):
> *Our results suggest that a space map is created by spatially organized localization cues. This is further supported by maps of sound localization cues in the auditory brainstem (Olsen et al., 1989; Carr & Konishi, 1990). Such maps of localization cues may explain why the cue-independent auditory cortex lacks a map of auditory space (Higgins et al., 2017). We find that the formation of a map does not benefit localization accuracy; instead, it tends to worsen accuracy in both models and human listeners (Fig. 7b vs. f; Fig. 8c vs. h). The auditory cortex may therefore trade the benefits of forming maps (minimizing wire cost, Chklovskii & Koulakov (2004)) against localization accuracy.*
>
> We look forward to your feedback on our global rebuttal and revised manuscript.

---

> > ### Comment · Reviewer_i53z · 2025-11-27
> > **Clarification**
> >
> > Thank you for your response and for your engagement with the rebuttal process.
> >
> > I've read all the reviews, rebuttals, and the updated text.
> >
> > The review raise 2 main concerns: one ragarding the model's architecture and another one regarding the parallels with biology. In response, the Authors announce a biologically-guided architecture and a new dataset. However, the updats in the manuscript (highlighted in blue) only provide new results regarding a 10-layer network (as opposed to a five-layer network), keeping the raised concerns in place.
> >
> > Please guide me where to look for the updated results as outlined above.
> > For now, as I couldn't locate those, I maintain my current valuation (but stay open to potential further updates)

---

> ### Author Response · Authors · 2025-11-27
> **Updated results in the manuscript**
>
> Thank you for reading our rebuttals and for the prompt responses. All the content after L268 is the updated text and figures on the human results, and we have now highlighted all of them in blue.

---

### Official Review · Reviewer_2GM3 · 2025-10-30

**Soundness:** 3
**Presentation:** 3
**Contribution:** 2
**Rating:** 4
**Confidence:** 4

**Summary:**

This paper investigates the existence of "what" and "where" processing streams in the auditory system and the question of the missing auditory space map. The authors train a 5-layer CNN solely on a sound localization ("where") task and analyze the learned representations using several quantitative and visualization methods. The results show that representations of sound type ("what") naturally emerge during localization training, suggesting that "what" and "where" information may be multiplexed in auditory processing. Furthermore, the spatial representations in the final layer organize into "maps," "patches," or "random patterns." However, these organizational patterns (e.g., map formation) show no significant correlation with localization performance. This finding implies that a topographic space map is not a necessary condition for accurate sound localization in the auditory system.

**Strengths:**

1. The analysis is comprehensive, employing multiple quantitative metrics (e.g., NMI, MSE, R²) to support the conclusions.
2. The visualizations of layer-wise representations across sound attributes (type, level, location) and microphone pairs are clear and informative.
3. The paper provides a compelling computational account that addresses two long-standing questions in auditory neuroscience: the coexistence of dual processing streams and the absence of topographic maps.

**Weaknesses:**

1. Although the 5-layer 1D CNN is inspired by the auditory pathway and takes raw binaural waveforms as input, the model lacks key characteristics of the biological auditory system and thus strong biological plausibility. Consequently, it remains unclear whether the emergent representations reflect genuine properties of the auditory system or are artifacts of the CNN architecture.
2. As acknowledged in the discussion, the paper’s key conclusions rely heavily on 2D UMAP embeddings, which are sensitive to hyperparameter choices. This represents a critical methodological weakness.
   - The R² metric (for the map-vs-performance claim) is computed on the 2D UMAP projections rather than the original high-dimensional representations.
   - It is also unclear whether the NMI metric (for the “what” clustering claim) is similarly derived from these 2D embeddings.

**Questions:**

1. Have the authors conducted analogous experiments on a visual "where" task (e.g., object localization)? If similar "what" representations emerge, it would suggest that such representational emergence may be a general property of hierarchical CNNs rather than an auditory-specific phenomenon.
2. Have the authors performed the reverse control experiment—training the model solely on a "what" task and testing whether "where" representations also emerge? Such an experiment would provide further evidence for the proposed multiplexing of "what" and "where" information.

---

> ### Author Response · Authors · 2025-11-27
> **Auditory system+CNN architecture+UMAP hyperparameter**
>
> We appreciate this reviewer for the constructive and insightful review.
>
> 1. “model lacks key characteristics of the biological auditory system.”\
>    We agree, and we believe the new human results are sufficient to address this concern. Please see the revised manuscript and the summary in the global rebuttal.
>
> 2. “artifacts of the CNN architecture.”\
>    A specific architecture (the 5-layer 1D CNN) is not critical, because we observed consistent findings in a 10-layer 1D CNN and in ten 2D CNN models with very different architectures. Therefore, we removed the claim that the “5-layer 1D CNN is inspired by the auditory pathway” due to the lack of strong evidence and further quantitative analysis. In the human results: 1) We showed 2D UMAPs from three architectures in Figs. 6b, 6d, 7d, 7e, 8d, 8e, and 8f; 2) We presented quantitative results from all ten architectures in Figs. 7f, 7g, 8g, 8h, and Supplementary Fig. 10; 3) We showed 2D UMAPs (and two metrics in figure legend) from all ten architectures in Supplementary Figs. 7, 9, 11, and 12. Together, our findings are consistent across architectures, both qualitatively and quantitatively.
>
> 3. "2D UMAP embeddings, which are sensitive to hyperparameter choices".\
>    This is an important point, and we kept it in mind during the rebuttal. The main hyperparameter of UMAP is the number of neighbors, and we have varied it in all new analyses, including the ten-layer 1D CNN and the human results. 1) In Figs. 8g and 8h (human spectral cue), we tested two values (20 and 50); 2) In Supplementary Figs. 5 and 6 (ten-layer 1D CNN), we tested seven values (10, 20, 50, 100, 200, 500, and 1000) and focused on five; 3) In Supplementary Fig. 8 (human reverberation), we tested six values (10, 20, 50, 100, 200, and 500); 4) In Supplementary Fig. 10 (human bandwidth), we tested three values (50, 100, and 200). Together, our findings are consistent across UMAP hyperparameter choices, both qualitatively and quantitatively.
>
> 4. "R² and NMI metrics are computed on the 2D UMAP".\
>    We chose these two metrics to quantify the organization strength and clustering accuracy of the 2D neural representations shown in nearly all Main and Supplementary Figures. In our original 5-layer 1D CNN, we also computed R2 in the original high-dimensional (i.e., 512D) representations. The 512D R2 scores are higher than the 2D R2 scores when the 2D R2 scores are very low. For example, in Supplementary Figs. 3a and 3b, the 512D/2D R2 scores are 0.97/0.99 and 0.96/0.22, respectively. In Supplementary Fig. 3e, six conditions have an R2 score less than 0.6 in 2D UMAP, but the minimal R2 score in 512D space is 0.89. The NMI metric is very stable across different dimensionalities, since the clustering is stable under manifold changes. Together, we could use the 512D metrics, but we do not yet know how to interpret those results in an intuitive way.
>
> 5. "visual 'where' task".\
>    This is an inspiring question. Although we have run out of time to test it, the answer is likely no. From a neuroscience perspective, the brain does not need to localize visual and somatosensory stimuli in the same way as audition because they are already mapped onto the retina and body, similar to how sound frequency is mapped in the cochlea. Different stimuli such as dots/bars/shapes/faces at the same location will activate specific neurons if they fall inside their spatial receptive fields (at least in V1/V2). In contrast, auditory spatial receptive fields and sound localization behavior are content-dependent from the beginning: spectral cues in DCN, low-frequency ITDs in MSO, and high-frequency ILDs in LSO. We believe that what and where pathways exist in all three sensory modalities (we have revised our previous claim that “what” is unnecessary), but only in audition does the where pathway have to carry the what. Therefore, we believe the emergence of what in where is an audition-specific phenomenon. This uniqueness of audition, rather than repeating what is already known in vision, is particularly interesting to the neuroscience–AI community.
>
> 6. "training the model solely on a 'what' task and testing whether 'where'".\
>    We would like to pursue this in the next step. This is an important question, and we have mentioned it in our Discussion (L531–533): “If DNN models are trained to perform purely ‘what’ tasks or combined ‘what’ and ‘where’ tasks (Saddler et al., 2025), the representation of ‘what’ should be even clearer than in ‘where’-only task-optimized models.”

---

### Official Review · Reviewer_c6YQ · 2025-10-31

**Soundness:** 2
**Presentation:** 2
**Contribution:** 2
**Rating:** 2
**Confidence:** 5

**Summary:**

The paper trains a simple neural network for sound source localisation. The network is composed of five 1D convolutional layers and presented with two microphone signals. The main contribution of the paper is that the network - trained for localisation - learns representations that are organised in some layers by sound category.

**Strengths:**

Source localisation depends on perceptual cues, such as interaural time and level differences. These cues are signal dependent. Therefore, I would expect intuitively that knowledge of the type of sound event benefits source localization. The results suggest that the specific model that is used in this paper leverages the category of sound events when optimizing the network for localisation.

**Weaknesses:**

My main concerns are 1) the validity and evidence to support claims and 2) the generality of the results. Regarding 1), the paper claims that the results evidence for a highly debated claim that the human auditory process can be separated into parallel pathways. However, as the authors acknowledged in the discussion in Section 4, a simplistic 5-layer 1D CNN is used for evaluation which bears little resemblance to the architecture of the human auditory process. As such, it is unclear to what extent the results support claims of parallel pathways in the human auditory process. 2) It is unclear if the conclusions are simply an artifact arising from the specific model architecture, or a phenomenon that is to be expected regardless of the model architecture. I would have expected to see a thorough evaluation across different architecture and hyperparameter settings. Considering that the architecture is largely unmotivated within the context of the vast literature on auditory modelling,  the results do not provide conclusive evidence of the claim that localisation – in general - benefits from sound event classification.

**Questions:**

To evidence the paper’s claim that  the results offer “a concrete computational account of multiplexed coding”:
-	Can you provide evidence that source localization models – in general – “replicate what and where representations”? Or are the results artifacts of the specific and simplistic architecture of the proposed model?
-	Can you provide evidence that the model architecture accurately captures the stages involved in the human auditory process?

---

> ### Author Response · Authors · 2025-11-27
> **Generality and resemblance to human auditory system**
>
> We appreciate this reviewer for raising these two important concerns, which inspired (and pushed) us to analyze neural representations in models of human sound localization behavior. One point we have deliberately toned down is the proposed correspondence between layers or stages of the model and specific brain regions in the auditory pathway. We do not yet have enough evidence or time to further pursue this important question during the rebuttal period.
>
> In our revised manuscript, we added a ten-layer model (L254–267, blue fonts; Supplementary Figs. 4–6) alongside our original five-layer model. While this addition alone is not sufficient, we believe that the new human results with ten localization–task-optimized models substantially alleviate concerns about model architecture.
>
> The neural representations in models of human sound localization show many resemblances to human auditory system, including:
>
> 1. maps are formed in both the horizontal and vertical planes when sounds contain binaural and monaural localization cues that are topographically organized relative to the human ears;
> 2. a space map is created by spatially organized localization cues, consistent with maps of sound localization cues in the auditory brainstem;
> 3. such maps of localization cues may explain why the cue-independent auditory cortex lacks a map of auditory space; and
> 4. formation of a map does not benefit localization accuracy. The auditory cortex, which does not have a map, may therefore trade the benefits of forming maps (minimizing wire cost) against localization accuracy.
>
> This rebuttal is short because we have already addressed the concerns about the generality of the models (architectures and hyperparameters) in the revised manuscript and summarized them in the global rebuttal. Please let us know your thoughts on this revised manuscript.

---

### Official Review · Reviewer_gYU6 · 2025-11-02

**Soundness:** 3
**Presentation:** 3
**Contribution:** 3
**Rating:** 4
**Confidence:** 4

**Summary:**

The work tries to address a neuroscience problem: does the auditory cortex need a "what" pathway and a "where" pathway, like in the visual cortex? The work investigates the problem from a computational perspective: construct a DNN to localize the sound sources and analyze whether the sound type and level representations could emerge in the DNN layers. The experiments show that the answer is yes, and it is concluded that the two pathway-structure is not necessary in the auditory cortex.

**Strengths:**

1. The work studies an interesting problem in neuroscience with the aid of DNN.
2. The presentation is clear in general though there are some minor problems.
3. The proposed method --- train a DNN to do sound localization only but analyze whether the DNN can develop representations of the sound identity (type and level) -- is novel and brave. People usually don't expect to obtain a positive answer to this question.

**Weaknesses:**

1. The logic behind the main conclusion seems to be problematic: A DNN trained to do a task of "where" develops the representation of "what", does not indicates that a separate "what" pathway is unnecessary in the auditory cortex. It is possible that this representation of "what" is just a byproduct of the task of "where". In other words, with a separate "what" pathway in the cortex, which corresponds to train the DNN to do tasks of "what", maybe the representation of "what" is clearer and/or the performances of "what"-tasks are better.

2. An explanation of why the representation of "what" emerges in the DNN trained to do the task of "where" is needed. This is a quite strange result, and an in-depth investigation is needed to consolidate the result.

3. To strengthen the conclusion, a symmetric experiment is needed: train a DNN to do a task of "what" and examine whether the model develop representations of "where".

4. A deeper network is needed to strengthen the result.

5. The sound location representations are presented as visual results only, but in many figures (e.g., the 5 subfigures in Fig. 5c), it is hard to tell their difference from visual inspection. A metric is needed here to quantify the results.

6. A large part of the paper (Figs. 3-7) is discussing whether the representation of "what" or "where" forms a map or patches. Do you assume that the map- or patch-form of pattern is a must for the existence of a pathway? Why shouldn't a pathway have a random pattern? Please clarify.

**Questions:**

All main questions are stated in the Weakness session. a minor question:
I don't understand L322-323: "Among all the six microphone pairs, only microphone pairs M12 and M24 form clusters for the sound type of white noise (Fig. 7)." What is special in the M12 and M24 subfigures? In addition, Fig. 7 caption doesn't say that the results were obtained with the sound type of white noise.

---

> ### Author Response · Authors · 2025-11-27
> **Logic+explanation+deeper+quantify**
>
> We thank this reviewer for the positive comments (novel and brave) on our work.
>
> 1. "does not indicates that a separate 'what' pathway is unnecessary".\
>    This is a great point and we agree. In the last sentence of the Abstract, we have rephrased “auditory cortex does not need to dissociate what and where” to “what cannot be dissociated from where in auditory cortex.” We further mention this in our Discussion (L527–531): “Our results do not imply that a separate ‘what’ pathway is unnecessary...”
>
> 2. "why the representation of 'what' emerges in the DNN trained to do the task of 'where' is needed".\
>    Figs. 7 and 8 in our revised manuscript largely address this question by manipulating the bandwidth and center frequency of simple sounds (instead of animal vocalizations or human speech). The “where” models relied on monaural and binaural localization cues, which are essentially “what” features.
>
> 3. "a symmetric experiment is needed: train a DNN to do a task of 'what'".\
>    This is also a good point, and we mention it in our revised Discussion (L531–533): “If DNN models are trained to perform purely ‘what’ tasks or combined ‘what’ and ‘where’ tasks (Saddler et al., 2025), the representation of ‘what’ should be even clearer than in ‘where’-only task-optimized models.”. We would absolutely add these experiments if we had one extra week of rebuttal time and additional pages. Due to the addition of the human results, we now have 8 Figures and 12 Supplementary Figures. We plan to pursue this in our next step.
>
> 4. "A deeper network is needed".\
>    We built a ten-layer DNN to confirm our findings (L254–267, blue fonts; Supplementary Fig. 4-6). The findings are consistent with the five-layer DNN. Stronger evidence comes from our human results, where 1500 models were optimized and we analyzed the representations in the top ten models. These human results are also consistent with our main findings.
>
> 5. "A metric is needed here to quantify the results".\
>    We used two metrics, clustering accuracy (NMI) and organization strength (R2), in all of the newly added figures. Adding these metrics allows us to compare the results with human behaviors quantitatively (Figs. 7, 8). The clustering accuracy is correlated with the models’ and humans’ accuracy, but organization strength is not.
>
> 6. "map- or patch-form of pattern is a must".\
>    This is an interesting point. We of course do not assume that a map- or patch-form pattern is required for the existence of a pathway. Map- or patch-form patterns are widely used in fMRI studies due to their low spatial resolution. If the pattern is totally random, fMRI cannot reveal any organization, e.g., in the study “Widespread and opponent fMRI signals represent sound location in macaque auditory cortex” (Ortiz-Rios et al., *Neuron*, 2017). Our models are best suited to explain fMRI results rather than single-neuron results in nonhuman primates.
>
> Minor question: "What is special in the M12 and M24 subfigures? Fig. 7 caption".\
> Thank you for raising this. The main difference is the inter-microphone distance: M12 is width, and M24 is diagonal. The “white” noise sound types are well segregated in all models but are twisted in some of them. We trained the model with all sound types instead of white noise alone, thus the MSE is much larger than when training on white noise only. We have removed original Fig. 7 due to space limitations and only mention M13 and M24 in the revised manuscript.

---

### Author Response · Authors · 2025-11-27
**Global rebuttal: new models and datasets from human sound localization behavior**

We truly appreciate the insightful comments from the four reviewers and the relatively positive scores for soundness (3233), presentation (3234), and contribution (3222). The common criticism from all reviewers is that only one simple 5-layer CNN is used in the study. Two reviewers who rejected this paper (c6YQ and i53z, rating: 2) also criticized the lack of connection between our model and human auditory processes/system.

To address the limitations in our models and datasets, we analyzed the neural representations of sound locations in models optimized for human sound localization behavior (Francl & McDermott, 2022, *Nature Human Behavior*; Saddler & McDermott, 2024, *Nature Communications*). These task-optimized models (10 out of 1500 candidate models) received sound inputs that were already filtered by the pinnae, head, and torso, and further filtered by the auditory nerve in the human cochlea. **Since these models exhibit many features of human spatial hearing, analyzing their neural representations allows us to bridge the gap between human sound localization behavior and the human auditory system, and to align our findings with human behavior.** Note that we could not find open-source human auditory cortex data on sound localization.

In the revised manuscript, the human results (starting at L268, blue) were added in Figs. 5–8 and Supplementary Figs. 7–12. We also revised the Abstract (blue), our contributions (L123, blue), and most of the Discussion (L515, blue). We **reproduced our main finding** (emergence of “what” clusters in “where” models), even though the models and datasets are totally different from the previous ones. There are **two new findings**: 1) formation of an auditory space map worsens localization accuracy in both models and human listeners (Fig. 7b vs. 7f; Fig. 8c vs. 8h); 2) space maps were formed when sounds contained localization cues that were topographically organized (Fig. 7c vs. 7e; Fig. 8b vs. 8f).

---

### Meta-Review · Area_Chair_LTWK · 2026-01-07

**Summary:**

The paper presents a CNN model that claims to mimic and possibly explain human auditory processing for localization. The reviewers were concerned that the conclusions about relevance of this model for human auditory processing and that the “what/where” pathways were overstated given the lack of biological realism and direct neural data. In term of quality of presentation and originality of the work, the reviewer were overall positively impressed, but methodological concerns and missing experiments (e.g., training on “what” only) seem to fall below the acceptance standards. Overall it seems that the lack of confidence in the generality and biological relevance of the results was shared by all reviewers

**Reviewer Concerns:**

The rebuttal addressed architectural generality by reviewing additional models networks and softening and clarifying the biological claims. The authors presented new findings about formation of an auditory space map, however, key issues remain, including the absence of additional control experiments and direct links to neural recordings in the auditory cortex limits the biological validity of the claims.

**Reviewer Scores:**

The two reviewers who were already borderline would likely have shifted to weak accept given the added models and reframed claims, while the two rejecting reviewers were very confident in their reject recommendation based on lack of control experiments, something that the rebuttal did not provide. One of the reviewers responded to the authors stating that the response was insufficient and also that he could not track the new results. It seems that the authors did not further respond to the reviewer inquery . Overall, even if the discussion might have converged on a slightly higher score, it would still likely be below-threshold because of questionable methods and implications for auditory neuroscience.

---

### Decision · Program_Chairs · 2026-01-26

Reject